# LightningRL: Breaking the Accuracy–Parallelism Trade-off of Block-wise dLLMs via Reinforcement Learning

Yanzhe Hu [1 2]   Yijie Jin [1]   Pengfei Liu [1]   Kai Yu [1]   Zhijie Deng [1]

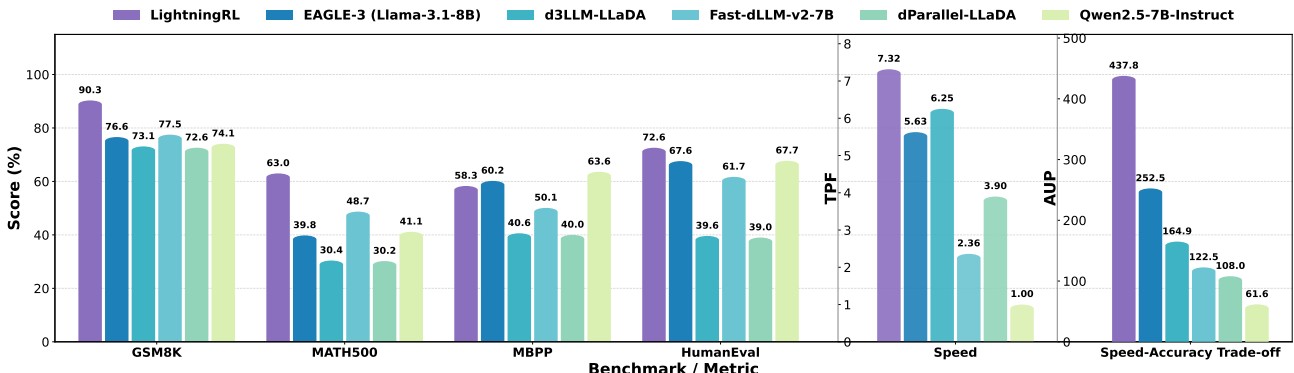

*Figure 1.* **Evaluation results of LightningRL and the baselines.** LightningRL achieves superior accuracy on math and code benchmarks while accelerating parallel decoding to an average of 7.32 tokens per forward (TPF) and 437.8 accuracy under parallelism (AUP) (Qian et al., 2026), significantly outperforming baselines including Eagle-3 (Li et al., 2025) and Fast-dLLM-v2 (Wu et al., 2025a).

## Abstract

Diffusion Large Language Models (dLLMs) enable parallel token generation, and their block-wise variants have attracted significant attention. However, existing dLLMs usually exhibit an accuracy–parallelism trade-off, where raising tokens per forward (TPF) via aggressive parallel decoding often degrades task accuracy. To address this, we suggest developing a post-training approach to directly optimize the speed–quality frontier of pre-trained dLLMs. Conceptually, we do not require the model to decode aggressively along all sampling trajectories, but rather to find several highly parallelizable ones that can yield correct results. To this end, we resort to a reinforcement learning paradigm, i.e., LightningRL, to optimize rewards regarding both the final accuracy and inference parallelism. LightningRL follows the Group Relative Policy Optimization (GRPO) framework,

with further improvements for dLLMs: 1) stabilized training via per-reward decoupled normalization, 2) token-level negative log-likelihood (NLL) loss on correct trajectories for regularization, and 3) improved training efficiency through dynamic sampling with TPF-aware filtering. Across maths and code tasks, LightningRL consistently advances the Pareto frontier, maintaining competitive accuracy while increasing parallelism to an average TPF of 7.32 (up to 11.10 on MBPP). The code is now available at https://github.com/SJTU-DENG-Lab/LightningRL.

## 1. Introduction

Diffusion Large Language Models (dLLMs) are a promising alternative to autoregressive (AR) decoders for high-throughput generation (Li et al., 2022; Sahoo et al., 2024a; Gulrajani & Hashimoto, 2023; Gong et al., 2023; Sahoo et al., 2024b; Nie et al., 2025; Ye et al., 2025). They frame the generation as iterative denoising of a corrupted token sequence, enabling bidirectional context aggregation and parallel refinement over many positions. However, vanilla dLLMs can suffer from the incompatibility with KV cache mechanisms and fixed generation length.

Work done during Yanzhe Hu's internship at Shanghai Jiao Tong University. [1]Shanghai Jiao Tong University [2]Huazhong University of Science and Technology. Correspondence to: Zhijie Deng <zhijied@sjtu.edu.cn>.

*Proceedings of the 43rd International Conference on Machine Learning*, Seoul, South Korea. PMLR 306, 2026. Copyright 2026 by the author(s).

Block-wise dLLMs (Arriola et al., 2025) address these by bridging AR and diffusion LLMs. They generate blocks of text tokens sequentially to ensure long-range coherence (AR-like), while simultaneously refining all tokens within each block via diffusion to unlock intra-block parallelism. This approach mitigates the inefficiency of independent token sampling in pure diffusion and the lack of parallelism in AR. In practice, representative works construct block-wise dLLMs by adapting pretrained AR models for block-wise denoising (Cheng et al., 2025; Wu et al., 2025a).

However, existing block-wise dLLMs still suffer from a severe accuracy–parallelism trade-off (Qian et al., 2026), where raising tokens per forward (TPF) via aggressive parallel decoding often degrades task accuracy. Although training-free sampling strategies (Wu et al., 2025b; Xu et al., 2025) and distillation-based approaches (Wang et al., 2025a; Chen et al., 2025) can boost the TPF and hence the tokens per second (TPS) during inference, this typically comes at the cost of substantial degradation in generation quality. As a result, existing dLLMs are still inferior to popular acceleration approaches to AR LLMs like speculative decoding (Leviathan et al., 2023; Li et al., 2025; Kou et al., 2024) when simultaneously considering speed and quality.

To address this, we advocate a post-training approach for pre-trained dLLMs that directly optimizes the speed–quality frontier. Our core insight is that we do not require the model to aggressively decode all possible paths; instead, we merely need the model to reliably navigate the specific subspace of trajectories that are both highly parallelizable and accurate. We formulate this objective as a reinforcement learning (RL) problem, using supervision from outcome accuracy and overall TPF to shape the model's probability mass. We implement the resulting **LightningRL** upon the widely-used Group Relative Policy Optimization (GRPO) framework (Shao et al., 2024).

LightningRL makes necessary modifications to GRPO, including 1) *Decoupled Reward Normalization*, which addresses the scale discrepancy between accuracy and TPF rewards via independent normalization to ensure stable multi-objective optimization, 2) *Likelihood-Anchored Regularization*, where a token-level negative log-likelihood (NLL) objective computed on correct trajectories is leveraged to mitigate reward hacking and stabilize updates, and 3) *TPF-aware Filtering*, which selects prompts with sampling trajectories of diverse levels of parallelism to maintain distinct learning signals and improve sample efficiency.

Empirically, we perform LightningRL post-training on the representative block-wise dLLMs SDAR (Cheng et al., 2025) and validate on a comprehensive suite of math and code benchmarks. We report accuracy, TPF (as a measure of parallelism), and accuracy under parallelism (AUP) (Qian et al., 2026), which summarizes the speed–quality trade-off under parallel decoding. The results highlight a superior trade-off profile: our LightningRL-8B model with a block size of 32, tuned from the SDAR-8B, achieves 437.8 AUP with an average speed of 7.32 TPF (up to 11.10 TPF on MBPP). This performance substantially surpasses established acceleration baselines such as d3LLM (Qian et al., 2026) and EAGLE-3 (Li et al., 2025), demonstrating the ability of LightningRL to effectively break the accuracy–parallelism bottleneck of dLLMs.

## 2. Preliminaries

### 2.1. Diffusion Large Language Models (dLLMs)

Given an input prompt $q$, a dLLM generates the response as a Markov Decision Process (MDP) (Li et al., 2022). Ideally, this process produces a trajectory of intermediate states $\{x_0, x_1, \ldots, x_T\}$, where each $x_t \in (\mathcal{V} \cup \{[\text{MASK}]\})^L$ represents the partially decoded sequence at step $t$. Here, $\mathcal{V}$ denotes the vocabulary, $L$ is the sequence length, and $x_0$ is the initial fully masked state.

Let $p_\theta(\cdot \mid x_t, q)$ denote the generation distribution parameterized by the model, which effectively serves as the transition policy. The generation proceeds by iteratively sampling the next state based on the current context:

$$x_{t+1} \sim p_\theta(x_{t+1} \mid x_t, q). \tag{1}$$

This aligns with the Markov property, where the next state depends only on the current state $x_t$ and the condition $q$.

**Confidence-driven Decoding** (Wu et al., 2025b; Wang et al., 2025a) specifies the transition dynamics by allowing multiple tokens in $x_t$ to be accepted (decoded) in a single iteration if their prediction confidence exceeds a threshold, while rejected tokens are reset to [MASK] in $x_{t+1}$.

**Block Diffusion** Vanilla dLLMs suffer from the incompatibility with KV cache mechanisms and fixed generation length (Nie et al., 2025; Ye et al., 2025). Block-wise dLLMs (Arriola et al., 2025; Cheng et al., 2025) address these by partitioning the sequence $x$ into $B$ contiguous blocks $\{x^1, \ldots, x^B\}$, where blocks are generated sequentially while tokens within each block are decoded in parallel. Block-wise dLLMs can be efficiently adapted from pretrained AR LLMs, with SDAR (Cheng et al., 2025) and Fast-dLLM-v2 (Wu et al., 2025a) as popular examples.

### 2.2. GRPO for dLLMs

Group Relative Policy Optimization (GRPO) (Shao et al., 2024) has demonstrated remarkable effectiveness for the reinforcement learning (RL) of AR LLMs. Prior works have successfully adapted this paradigm to the non-autoregressive decoding of dLLMs (Wang et al., 2025b; Zhu et al., 2026). To align the RL objective with the dLLM generation process,

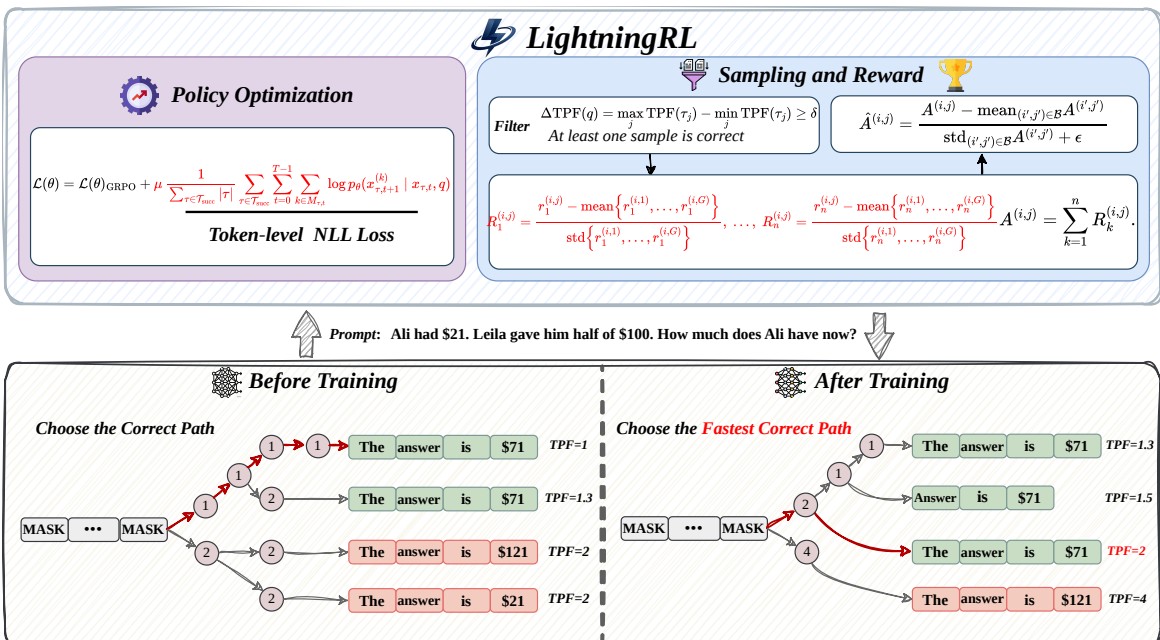

*Figure 2.* **Overview of LightningRL.** LightningRL samples a group of decoding trajectories per prompt, applies per-reward decoupled normalization to preserve within-group ranking under heterogeneous scales. The policy is optimized with a GRPO-style objective plus a token-level NLL anchor. The bottom panel shows the resulting shift toward the fastest correct trajectory, improving TPF without degrading accuracy.

we explicitly map the policy $\pi_\theta$ to the conditional denoising distribution $p_\theta(\cdot \mid x_t, q)$, and the trajectory $\tau$ to the sequence of intermediate noisy states $\{x_t\}$.

Specifically, for each prompt $q$, we sample a group of $G$ rolled-out trajectories $\{\tau_j\}_{j=1}^G$ using the behavior policy $\pi_{\theta_{\text{old}}}$. GRPO computes the terminal reward $R(\tau_j)$ and derives the advantage $\hat{A}_j$:

$$\hat{A}_j = \frac{R(\tau_j) - \frac{1}{G}\sum_{i=1}^G R(\tau_i)}{\text{std}(\{R(\tau_i)\}_{i=1}^G) + \epsilon}, \qquad (2)$$

where $\epsilon$ is a small constant for numerical stability.

GRPO maximizes the advantage-weighted probability of valid actions. Substituting our dLLM-specific policy definition, the loss function is formulated as:

$$\mathcal{J}_{\text{GRPO}}(\theta) = \mathbb{E}\Bigg[\frac{1}{G}\sum_{j=1}^G \frac{1}{|\tau_j|}\sum_{t=0}^{T-1}\sum_{k \in M_{j,t}}\Bigg(\\ \min\Big(\rho_{j,t,k}\hat{A}_j,\ \text{clip}(\rho_{j,t,k}, 1-\epsilon, 1+\epsilon)\hat{A}_j\Big)\\ - \beta D_{\text{KL}}\big(p_\theta(\cdot|x_{j,t})\|p_{\text{ref}}(\cdot|x_{j,t})\big)\Bigg)\Bigg].$$

$$(3)$$

Here, $M_{j,t} = \{i \mid x_{j,t}^{(i)} = \texttt{[MASK]}\}$ denotes the set of indices in the intermediate state $x_{j,t}$ that require denoising,

consistent with the generation process defined in Eq. 1. Accordingly, $|\tau_j| = \sum_{t=0}^{T-1}|M_{j,t}|$ represents the total number of parallel token predictions performed along the trajectory. $\beta$ is the coefficient for the KL divergence penalty against the reference model $p_{\text{ref}}$. The importance ratio $\rho_{j,t,k}$ is:

$$\rho_{j,t,k} = \frac{p_\theta(x_{j,t+1}^{(k)} \mid x_{j,t}, q)}{p_{\theta_{\text{old}}}(x_{j,t+1}^{(k)} \mid x_{j,t}, q)}. \qquad (4)$$

The components marked in red highlight the structural differences from standard AR-LLMs — gradients are propagated through parallel masked positions over multiple denoising steps conditioned on the intermediate noisy states, rather than sequential token positions conditioned on history.

## 3. LightningRL: Breaking the Accuracy–Parallelism Trade-off

To push the frontier of dLLMs through RL, we initially adopted existing RL frameworks established for dLLMs (Zhao et al., 2025; Wang et al., 2025b; Zhu et al., 2026). However, we encountered significant challenges due to our multi-objective setting (i.e., simultaneously optimizing for accuracy and speed). We observed reward collapse, where one objective dominates the optimization. Thus, the policy drifts, and the model's generation capability degrades.

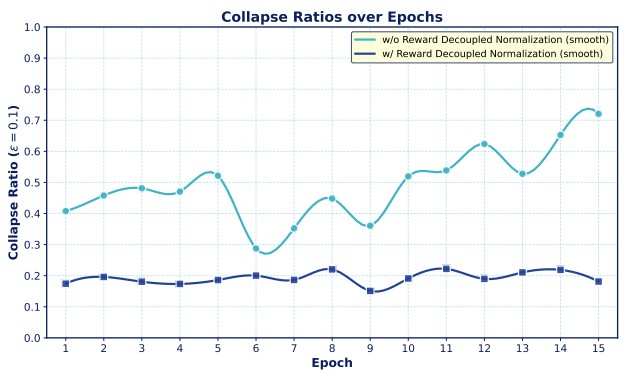

*(a)* Collapse ratio across epochs on GSM8K (lower is better).

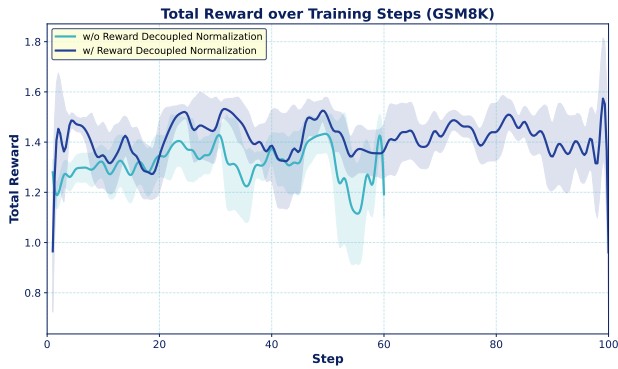

*(b)* Total reward during training on GSM8K.

*Figure 3.* **Per-reward decoupled normalization improves training stability.** It reduces signal collapse (a) and yields more stable reward optimization (b) under the same training setup.

To address these, we present LightningRL, a robust training algorithm designed to co-optimize accuracy and inference parallelism. The overall architecture of LightningRL is illustrated in Fig. 2, which incorporates three specific modifications designed for the dLLM landscape:

- Per-reward Decoupled Normalization (Sec. 3.1), which mitigates reward collapse by independently normalizing distinct reward signals.

- Token-level NLL Regularization (Sec. 3.2), which is applied to correct trajectories to prevent policy drift and maintain linguistic coherence.

- Dynamic Sampling with TPF-aware Filtering (Sec. 3.3), which helps efficiently explore the trade-off between speed and quality.

### 3.1. Decoupled Normalization for Group Rewards

In our formulation, each rollout $\tau^{(i,j)}$ receives two terminal reward signals: a correctness reward $r_{\mathrm{acc}}^{(i,j)} = c^{(i,j)}$, where $c^{(i,j)}$ is the verifier-based correctness indicator, and a speed reward $r_{\mathrm{tpf}}^{(i,j)} = \mathrm{TPF}(\tau^{(i,j)})$, which measures the decoding parallelism of the trajectory.

While GRPO is effective for single-objective optimization, a naive extension to multi-reward settings, which involves summing raw rewards and subsequently applying standard group-wise normalization, tends to be suboptimal. When a coarse, high-magnitude discrete reward (e.g., $r_{\mathrm{acc}} \in \{-1, 1\}$) is combined with a fine-grained continuous reward (e.g., $r_{\mathrm{tpf}}$), the aggregate value is frequently dominated by the discrete term, resulting in numerous within-group ties or near-ties. This phenomenon renders advantages indistinguishable across different behaviors and consequently diminishes the strength of the effective policy-gradient signal.

To mitigate this issue, we propose per-reward decoupled normalization. Instead of normalizing the aggregated reward, we normalize each reward component independently within the group and then aggregate the standardized signals. Consider the $i$-th prompt in the current minibatch and its generated group of rollouts of size $G$. For the $j$-th rollout, let $r_k^{(i,j)}$ denote the raw value of the $k$-th reward objective among a total of $n$ objectives. Instead of summing $r_k$, we first compute the independent advantage for each objective:

$$R_k^{(i,j)} = \frac{r_k^{(i,j)} - \mathrm{mean}\left\{r_k^{(i,1)}, \ldots, r_k^{(i,G)}\right\}}{\mathrm{std}\left\{r_k^{(i,1)}, \ldots, r_k^{(i,G)}\right\} + \epsilon}, \quad (5)$$

where $\epsilon$ is a small constant for numerical stability. This ensures that every reward component contributes an equally standardized ranking signal, regardless of its raw magnitude or distribution. In our setting with $n = 2$, we instantiate Eq. 5 for the accuracy and speed rewards respectively:

$$A_{\mathrm{acc}}^{(i,j)} = \frac{r_{\mathrm{acc}}^{(i,j)} - \mu_{\mathrm{acc}}^{(i)}}{\sigma_{\mathrm{acc}}^{(i)} + \epsilon}, \qquad A_{\mathrm{tpf}}^{(i,j)} = \frac{r_{\mathrm{tpf}}^{(i,j)} - \mu_{\mathrm{tpf}}^{(i)}}{\sigma_{\mathrm{tpf}}^{(i)} + \epsilon}, \quad (6)$$

where $\mu_{(\cdot)}^{(i)}$ and $\sigma_{(\cdot)}^{(i)}$ denote the within-group mean and standard deviation of each reward for the $i$-th prompt. The composite advantage is then obtained as $A^{(i,j)} = A_{\mathrm{acc}}^{(i,j)} + A_{\mathrm{tpf}}^{(i,j)}$. Finally, we apply global batch-wise normalization to control the update scale regardless of the number of objectives, yielding the final advantage $\hat{A}^{(i,j)}$ used for policy updates:

$$\hat{A}^{(i,j)} = \frac{A^{(i,j)} - \mathrm{mean}_{(i',j') \in \mathcal{B}} A^{(i',j')}}{\mathrm{std}_{(i',j') \in \mathcal{B}} A^{(i',j')} + \epsilon}, \quad (7)$$

where $\mathcal{B}$ denotes the set of all rollout indices $(i', j')$ in the current minibatch. Crucially, while the preceding decoupling step preserves group-relative distinctions contributed by each objective, this final normalization prevents the advantage magnitude from drifting with the reward dimensionality $n$.

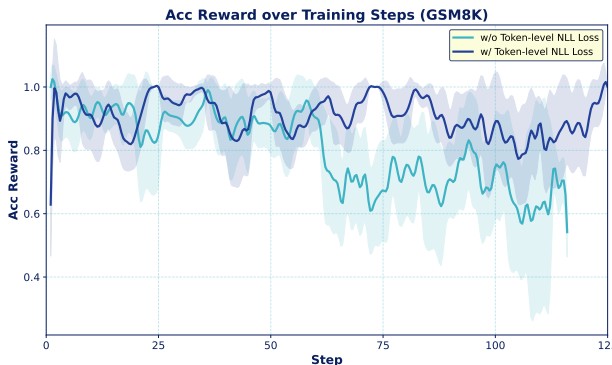

*Figure 4.* **Token-level NLL loss anchors the accuracy objective on GSM8K.** Compared with training without the token-level NLL term, it maintains a higher accuracy reward and mitigates late-stage drift in the accuracy signal under the same setup.

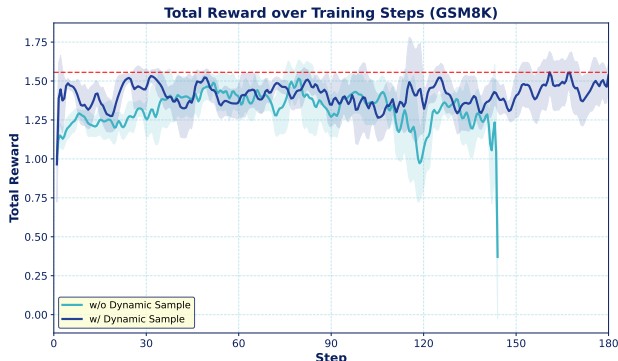

*Figure 5.* **Dynamic sampling improves robustness of total-reward optimization on GSM8K.** Relative to training without dynamic sampling, it reduces reward collapse under the same setup.

To validate our motivation, we present two diagnostics. First, Fig. 3a shows that decoupled normalization substantially reduces the Collapse Ratio, defined as the proportion of within-group pairs with advantage differences below $\epsilon$, thereby preserving finer preference resolution. Second, as shown in Fig. 3b, removing this decoupling step leads to noisier optimization trajectories, slower convergence, and lower peak performance. These results confirm that decoupling improves the conditioning of aggregated advantages by maintaining meaningful distinctions while stabilizing the update scale.

### 3.2. Token-Level NLL for Accuracy Anchoring

In multi-objective RL with verifier-based supervision, rewards on math and code tasks are typically sparse and sequence-level. In this setting, the policy gradient estimator necessarily takes a score-function form, where a scalar advantage multiplies log-probability gradients of the actions taken along the trajectory.

Formally, GRPO assigns each sampled trajectory $\tau$ a single group-relative advantage $\hat{A}(\tau)$. Abstracting away the piece-wise clipping details, the induced policy update admits the canonical score-function structure:

$$\nabla_\theta \mathcal{L}_{\text{Policy}} \propto -\mathbb{E}\left[\frac{\hat{A}(\tau)}{|\tau|} \sum_{t=0}^{T-1} \sum_{k \in M_{\tau,t}}\right.$$
$$\left. \nabla_\theta \log p_\theta(x_{\tau,t+1}^{(k)} \mid x_{\tau,t}, q)\right] \quad (8)$$

Crucially, $\hat{A}(\tau)$ is shared by all decoding steps in a trajectory: the factor $1/|\tau|$ only rescales the update and does not refine credit assignment. In multi-objective training, the *accuracy-driven* gradient can be substantially attenuated

by per-token dilution and further overshadowed when non-accuracy rewards dominate the effective advantage. This weakens the corrective pressure toward correctness and can induce drift into fast-but-incorrect modes. This motivates a positive-example LM loss that turns verifier-correct trajectories into dense, token-factorized anchoring toward correct behaviors.

To maximize the utility of rare verified successes and explicitly anchor the policy toward correctness, we introduce a token-level NLL loss that converts sequence-level successes into dense token-factorized supervision. Let $\mathcal{T}_{\text{succ}}$ denote the set of verifier-correct trajectories among on-policy samples in the current update. We define:

$$\mathcal{L}_{\text{NLL}}(\theta) = -\frac{1}{\sum_{\tau \in \mathcal{T}_{\text{succ}}} \sum_{t=0}^{T-1} |M_{\tau,t}|} \sum_{\tau \in \mathcal{T}_{\text{succ}}}$$
$$\sum_{t=0}^{T-1} \sum_{k \in M_{\tau,t}} \log p_\theta(x_{\tau,t+1}^{(k)} \mid x_{\tau,t}, q), \quad (9)$$

and set $\mathcal{L}_{\text{NLL}}(\theta) = 0$ when $\mathcal{T}_{\text{succ}} = \emptyset$. Here, consistent with Sec. 2.1, $M_{\tau,t}$ denotes the masked indices at step $t$ for trajectory $\tau$. Overall, we optimize the combined objective:

$$\mathcal{L}_{\text{LightningRL}}(\theta) = \underbrace{\mathcal{L}_{\text{Policy}}(\theta) + \beta\,\mathcal{L}_{\text{KL}}(\theta)}_{\mathcal{L}_{\text{GRPO}}(\theta)} + \mu\,\mathcal{L}_{\text{NLL}}(\theta),$$
$$(10)$$

where $\beta$ and $\mu$ are scalar coefficients balancing the loss components. This formulation yields a clean division of labor: $\nabla_\theta \mathcal{L}_{\text{GRPO}}$ drives preference learning via relative ranking across sampled trajectories, while $\mu \nabla_\theta \mathcal{L}_{\text{NLL}}$ acts as a self-imitation anchor that reallocates probability mass onto verifier-correct trajectories with stable, token-factorized gradients. Fig. 4 supports this anchoring effect: while training without token-level NLL loss exhibits a pronounced downward drift, the anchored variant maintains consistently higher accuracy rewards and substantially improved stability

over training steps.

### 3.3. Dynamic Sampling for Efficient Policy Optimization

Standard GRPO can suffer from gradient starvation under reward quantization. GRPO estimates advantages based on within-group relative rewards; consequently, when the $G$ sampled trajectories $\{\tau_j\}_{j=1}^G$ for a prompt $q$ fall into the same reward bin, the relative advantages vanish. This issue is particularly pronounced in our multi-objective setting: once the accuracy reward saturates within a group, differentiation relies solely on the speed reward, which is derived from discrete tokens per forward values. As a result, many groups yield identical total rewards $R(\tau_j)$, leading to vanishing gradients and wasted rollout budgets. Furthermore, if all samples within a group are incorrect, the total reward becomes entirely dominated by the speed reward, which often results in reward hacking.

To mitigate this issue, we propose dynamic sampling with TPF-aware filtering. We define the within-group TPF spread as:

$$\Delta \text{TPF}(q) = \max_j \text{TPF}(\tau_j) - \min_j \text{TPF}(\tau_j) \geq \delta, \quad (11)$$

where $\delta$ is a predefined filtering threshold. For each candidate prompt, we first sample a group of trajectories and accept the prompt only if it satisfies the aforementioned criterion and at least one sample is correct; otherwise, we discard it and resample a new prompt. We repeat this accept–reject process until the batch is filled. By filtering out near-tie groups, this procedure keeps a more consistent fraction of non-zero advantages in each update, leading to denser and more stable policy gradient signals.

Empirically, Fig. 5 shows that without dynamic sampling the training curve converges more slowly, reaches a lower plateau, and eventually undergoes a pronounced collapse, whereas with dynamic sampling training remains stable and achieves faster, higher-reward convergence under the same configuration.

## 4. Experiments

### 4.1. Setup

**Model and Dataset** We implement LightningRL upon the SDAR model family (Cheng et al., 2025) due to its leading performance. We refer to the 8B SDAR model with a block size of 32 as SDAR-8B-b32, with all other variants named likewise. For training, we utilize the training split of the MATH (Hendrycks et al., 2021) dataset and GSM8K (Cobbe et al., 2021) for mathematical reasoning and the PrimeIntellect (Team et al., 2025) dataset for code generation tasks.

**Reinforcement Learning Settings** During data collection,

we use a batch size of 128 tasks and a group size of 32 responses per task. We employ a low confidence dynamic sampling strategy with a threshold $\phi = 0.9$ and a sampling temperature of 1.0. We provide additional training details in the Appendix B.1.

**Benchmark and Baselines** We evaluate on four representative benchmarks: GSM8K (Cobbe et al., 2021), MATH500 (Hendrycks et al., 2021), HumanEval (Chen et al., 2021), and MBPP (Austin et al., 2021). To ensure a fair comparison with prior work, we employ a 4-shot setting for MATH and a 3-shot setting for Llada-based models (Nie et al., 2025) on MBPP. All other evaluations are conducted in a zero-shot setting. We benchmark our approach against state-of-the-art dLLMs and AR models.

**Evaluation Metrics** We assess performance based on three key evaluation metrics: TPF for parallelism, accuracy, and the AUP (Qian et al., 2026) score. Furthermore, we detail the TPS results for our model in Appendix C.2.

### 4.2. Main Results

As demonstrated in Tab. 1, LightningRL consistently outperforms DiRL and TraceRL across four benchmarks, especially in the coding domain. On the MBPP dataset, LightningRL achieves an AUP of 412.9 and a TPF of 11.10. These results substantially surpass TraceRL, which scores 144.2 in AUP and 2.50 in TPF, as well as DiRL with a TPF of 2.70. Beyond code generation, LightningRL maintains a strong advantage in mathematical reasoning tasks. The framework records AUP scores of 507.5 on GSM8K and 409.2 on MATH500, confirming its broad effectiveness in accelerating convergence and improving overall training efficiency across diverse domains.

### 4.3. Training Dynamic

As illustrated in Fig. 6, compared to the baseline approaches, LightningRL effectively prevents deviations in the optimization direction across both mathematical and coding tasks. We observe that a deviating optimization process produces widespread negative advantages. While this reduces the probability of undesirable patterns, it concurrently suppresses favorable ones, leading to overall training failure. A comparison between the training curves of TraceRL and GRPO(traj) reveals that training stability improves significantly upon the removal of the value model. A further discussion of this phenomenon is provided in Appendix A, and the training dynamics are detailed in Appendix C.3.

### 4.4. Analysis of Decoding Behavior

Figure 7 illustrates the step-wise decoding dynamics of the model trained using LightningRL. In comparison to SDAR, LightningRL executes the decoding process across multiple

*Table 1.* **Evaluation results of LightningRL and methods on math and code benchmarks.** All methods are trained on the SDAR 8B model with block size 32. Compared with the other three algorithms and the SDAR base model, LightningRL delivers substantial improvements across both math and code benchmarks.

| Model | GSM8K | | | MATH500 | | | MBPP | | | HumanEval | | |
|---|---|---|---|---|---|---|---|---|---|---|---|---|
| | Acc (%) | TPF | AUP | Acc (%) | TPF | AUP | Acc (%) | TPF | AUP | Acc (%) | TPF | AUP |
| SDAR (Cheng et al., 2025) | 88.9 | 2.85 | 252.5 | **63.6** | 4.81 | 318.9 | 58.0 | 2.44 | 108.4 | 73.5 | 2.39 | 165.0 |
| Coupled-GRPO (Gong et al., 2025) | 75.3 | 4.22 | 254.7 | 59.1 | 4.93 | 283.1 | 54.0 | 2.61 | 139.9 | 68.2 | 2.42 | 164.6 |
| TraceRL (Wang et al., 2025b) | 76.9 | 5.04 | 378.6 | 60.6 | 4.82 | 284.6 | 57.8 | 2.50 | 144.2 | 75.0 | 2.29 | 171.6 |
| DiRL (Zhu et al., 2026) | 86.6 | 4.87 | 414.1 | 61.6 | 5.04 | 301.2 | 56.8 | 2.70 | 152.7 | **76.0** | 2.33 | 176.7 |
| **LightningRL** | **90.3** | **5.58** | **507.5** | 63.0 | **6.28** | **409.2** | **58.3** | **11.10** | **412.9** | 72.6 | **6.30** | **421.5** |

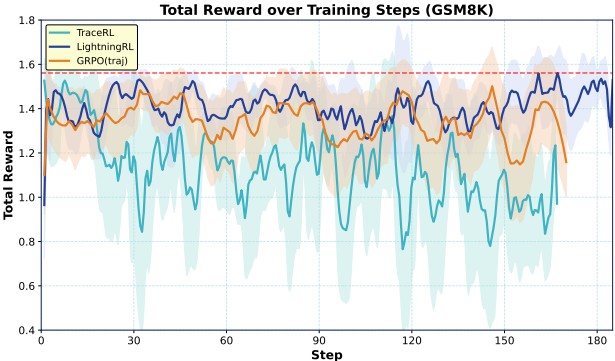

*Figure 6.* **Total Reward curves over training steps.** LightningRL outperforms baseline methods by suggesting a more reliable optimization landscape.

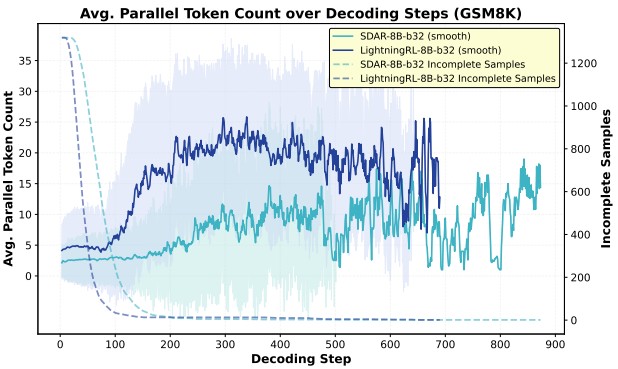

*Figure 7.* **Decoding behavior.** LightningRL maintains higher throughput and reduces long-tail decoding steps relative to the baseline.

samples with significantly higher parallelism. Specifically, the majority of samples processed by LightningRL terminate within approximately 100 steps, whereas the decoding process of SDAR exhibits a heavy-tail distribution extending beyond 850 steps. This improvement in performance stems from a higher throughput achieved during the active decoding phase. Consequently, LightningRL effectively mitigates the occurrence of long-tail decoding steps.

### 4.5. Benchmark Result

An illustration of the comparison between LightningRL and leading baselines is provided in Fig. 1, with a complete comparison with more baselines detailed in Tab. 2. As shown, LightningRL-8B-b32 achieves an average AUP of 437.8 and an average TPF of 7.32 on these baselines, with an average accuracy of 71.1%. In comparison, while yielding an almost identical average accuracy, the SDAR-8B-b32 model attains only 211.2 AUP and 3.12 TPF. Moreover, LightningRL consistently dominates representative AR and diffusion baselines, surpassing EAGLE-3 (Li et al., 2025) (252.5 average AUP, 5.63 average TPF) and the d3LLM (Qian et al., 2026) family across the same evaluation settings.

### 4.6. Ablation

**Ablation Study on three components.** To assess the contribution of each component within LightningRL, we conduct

ablation studies on the LightningRL-8B-b32 model using the GSM8K dataset. As shown in Tab. 3, we compare our full method against three variants. Empirical results confirm that all components are integral to LightningRL's peak performance of 90.3% accuracy, 5.58 TPF, and 507.5 AUP. Ablations lead to clear performance drops: removing the NLL loss causes the most substantial accuracy decline to 80.7%, omitting decoupled normalization reduces accuracy to 85.3% and TPF to 4.96, and excluding TPF-aware filtering lowers accuracy to 87.2%.

**Ablation Study on different loss reduction strategies.** LightningRL optimizes a weighted sum of policy, KL regularization, and NLL losses. We investigate loss reduction strategies by comparing token-level and sequence-level averaging. Token-level reduction weights all tokens globally, while sequence-level reduction averages within each sequence to weight rollouts equally. These choices induce distinct implicit sample weightings under varying sequence lengths, substantially altering training dynamics. As Tab. 4 demonstrates, the Seq-Tok-Tok configuration achieves an optimal accuracy and AUP, whereas pure token-level reduction degrades accuracy to 80.0%. Prior work (Ou et al., 2025) attributes this degradation to the mismatch between autoregressive objectives and non-autoregressive diffusion processes. Since diffusion models lack tractable token-level conditional likelihoods, current methods rely on heuristic

*Table 2.* **Evaluation results of LightningRL and the baselines on math and code benchmarks.** The notation *LightningRL-8B-b32* denotes the 8B LightningRL model with a block size of 32. LightningRL consistently advances the speed–quality frontier over its SDAR baseline and prior diffusion and autoregressive baselines, achieving substantially higher AUP at comparable accuracy.

| Model | GSM8K | | | MATH500 | | | MBPP | | | HumanEval | | |
|---|---|---|---|---|---|---|---|---|---|---|---|---|
| | Acc (%) | TPF | AUP | Acc (%) | TPF | AUP | Acc (%) | TPF | AUP | Acc (%) | TPF | AUP |
| *dLLMs* | | | | | | | | | | | | |
| Dream (Ye et al., 2025) | 83.9 | 1.00 | 83.9 | 39.6 | 1.00 | 39.6 | 57.2 | 1.00 | 57.2 | 55.2 | 1.00 | 55.2 |
| Fast-dLLM-Dream (Wu et al., 2025b) | 79.0 | 1.44 | 110.4 | 38.3 | 1.78 | 49.0 | 53.2 | 1.20 | 63.6 | 54.3 | 1.33 | 63.2 |
| dParallel-Dream (Chen et al., 2025) | 82.1 | 3.02 | 215.2 | 38.7 | 2.94 | 62.6 | 55.4 | 2.24 | 108.4 | 54.3 | 2.57 | 97.1 |
| d3LLM-Dream (Qian et al., 2026) | 81.4 | 4.94 | 333.6 | 38.2 | 3.92 | 74.6 | 55.6 | 2.96 | 142.1 | 57.1 | 3.20 | 126.6 |
| LLaDA (Nie et al., 2025) | 72.6 | 1.00 | 72.5 | 32.2 | 1.00 | 32.2 | 47.1 | 1.00 | 41.7 | 38.3 | 1.00 | 38.3 |
| Fast-dLLM-LLaDA (Wu et al., 2025b) | 74.7 | 2.77 | 153.7 | 30.8 | 1.97 | 38.8 | 38.6 | 2.13 | 56.6 | 37.8 | 2.56 | 52.1 |
| D2F-LLaDA (Wang et al., 2025a) | 73.2 | 2.88 | 158.6 | 28.7 | 2.38 | 38.5 | 38.0 | 1.94 | 53.0 | 36.6 | 2.69 | 59.2 |
| dParallel-LLaDA (Chen et al., 2025) | 72.6 | 5.14 | 246.7 | 30.2 | 3.17 | 46.6 | 40.0 | 2.35 | 60.5 | 39.0 | 4.93 | 78.1 |
| d3LLM-LLaDA (Chen et al., 2025) | 73.1 | **9.11** | 416.1 | 30.4 | 5.74 | 65.9 | 40.6 | 4.21 | 88.4 | 39.6 | 5.95 | 89.3 |
| *AR Models* | | | | | | | | | | | | |
| Qwen-2.5-7B-it (Qwen et al., 2025) | 74.1 | 1.00 | 74.1 | 41.4 | 1.00 | 41.1 | **63.6** | 1.00 | 63.6 | 67.7 | 1.00 | 67.7 |
| EAGLE-3 (LLaMA-3.1) (Li et al., 2025) | 76.6 | 5.12 | 276.5 | 39.8 | 5.72 | 100.9 | 60.2 | 5.69 | 300.7 | 67.6 | 5.98 | 331.9 |
| *Block-wise dLLMs* | | | | | | | | | | | | |
| Fast-dLLM-v2 (Wu et al., 2025a) | 77.5 | 2.21 | 158.7 | 48.7 | 2.61 | 90.5 | 50.1 | 2.04 | 114.6 | 61.7 | 2.58 | 126.1 |
| SDAR-8B-b32 (Cheng et al., 2025) | 88.9 | 2.85 | 252.5 | **63.6** | 4.81 | 318.9 | 58.0 | 2.44 | 108.4 | **73.5** | 2.39 | 165.0 |
| **LightningRL-8B-b32** | **90.3** | 5.58 | **507.5** | 63.0 | **6.28** | **409.2** | 58.3 | **11.10** | **412.9** | 72.6 | **6.30** | **421.5** |

*Table 3.* **Ablation study of LightningRL components on GSM8K.** We present the accuracy, parallelism, and AUP score of the three components at the highest AUP during 20 training epochs. The results show that each component is critical to maintaining the optimal Pareto frontier.

| Model | Acc (%) | TPF | AUP |
|---|---|---|---|
| w/o NLL loss | 80.7 | 5.03 | 385.7 |
| w/o Decoupled normalization | 85.3 | 4.96 | 416.5 |
| w/o TPF-aware filtering | 87.2 | 5.27 | 454.5 |
| **LightningRL** | **90.3** | **5.58** | **507.5** |

*Table 4.* **Ablation study of loss reduction strategies on GSM8K.** We present the accuracy, parallelism, and AUP score on different loss reduction strategies. The notation *Seq-Tok-Tok* denotes the reduction strategy of policy loss, KL loss, and NLL loss.

| Reduction Strategy | Acc (%) | TPF | AUP |
|---|---|---|---|
| Seq-Seq-Seq | 87.5 | 5.42 | 467.1 |
| Seq-Tok-Seq | 88.7 | 4.86 | 424.7 |
| **Seq-Tok-Tok** | **90.3** | **5.58** | **507.5** |
| Tok-Tok-Tok | 80.0 | 3.91 | 306.5 |

proxies that introduce bias and inconsistency during token-level optimization.

# 5. Related Work

## 5.1. dLLM Acceleration

dLLMs offer a non-autoregressive generation paradigm that iteratively denoises a fully or partially masked sequence, enabling parallel token updates but leaving practical inference efficiency still comparatively underexplored. Recent acceleration work largely falls into: (i) reducing per-step compute via diffusion-compatible caching and confidence-aware decoding rules (Wu et al., 2025b;a); (ii) increasing effective parallelism and/or cutting denoising steps through improved unmasking, sampling, and learned parallel decoding strategies (Xu et al., 2025; Bao et al., 2025; Chen et al., 2025); and (iii) hybrid diffusion–autoregressive pipelines that better leverage block-wise decoding and KV-cache-style components (Wang et al., 2025a). Our work builds upon these acceleration techniques but focuses on optimizing the decoding trajectory via reinforcement learning to mitigate the accuracy loss typically associated with high parallelism.

## 5.2. Reinforcement Learning in dLLMs

Recent advances in dLLMs leverage RL for step-wise denoising. Beyond early masked SFT and policy optimization (Zhao et al., 2025; Gong et al., 2025), research has shifted toward trajectory-level optimization, including step-aligned scheduling (He et al., 2025), trace-aware RL (Wang et al., 2025b; Zhu et al., 2026). Despite these gains, existing frameworks prioritize generation quality and relegate parallelism to a mere inference-time adjustment. LightningRL addresses this gap by treating parallelism as a first-class training objective, employing multi-objective RL to jointly optimize speed and accuracy. Decoupled and anchored regularization adopted by LightningRL stabilizes multi-objective training against sparse-reward instability.

# 6. Conclusion

In this paper, we presented LightningRL, an RL framework designed to optimize the speed–quality trade-off in diffusion language models. By mitigating the error amplification inherent in aggressive parallel decoding, LightningRL reconciles the conflict between generation speed and accu-

racy. Our results on maths and code generation benchmarks demonstrate that LightningRL consistently achieves higher AUP under high parallelism constraints, paving the way for practical, high-throughput dLLM deployment. Future work will explore scaling laws with larger contexts and generalize this approach to a wider range of dLLM generation tasks.

## Impact Statement

This paper presents foundational research in deep learning, aiming to advance the field of machine learning. While optimizing the accuracy–parallelism frontier and inference efficiency of Diffusion Large Language Models via reinforcement learning may yield potential societal impacts, we do not engage in a detailed discussion here, as the ethical implications and anticipated societal consequences of developments in efficient generative AI are already well-established.

## Acknowledgements

This work was supported by Shanghai Key Technology R&D Program "New Generation of Information Technology" (No. 25511103700), NSF of China (Nos. 62306176, 92470118), CCF-ALIMAMA TECH Kangaroo Fund (NO. CCF-ALIMAMA OF 2025010), and Ant Group.

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

# A. Discussion on Value Model Incorporation

| Method | Acc (%) | TPF | AUP |
|---|---|---|---|
| w/ Value Model | 80.7 | 5.14 | 408.1 |
| **LightningRL** | **90.3** | **5.58** | **492.4** |

*Table 5.* Comparison of Performance with and without Value Model Incorporation on GSM8K. The AUP in this table is computed using each model's own accuracy at TPF≈1 as $y_{\max}$.

To further assess whether a learned critic benefits LightningRL, we integrated a value network $V_\phi$ and used GAE for advantage estimation. As shown in Tab. 5, this variant yields a clear drop in both accuracy and TPF.

We conjecture that the critic becomes unreliable under the highly non-smooth state transitions induced by block-wise decoding, causing frequent *advantage sign flips*. Let $V_\phi(x_t) = V^\pi(x_t) + \epsilon(x_t)$, where $\epsilon(x_t)$ denotes the value approximation error. Under GAE, the estimated advantage can be written as the true advantage plus an accumulated error term that is a weighted combination of $\epsilon(x_{t+l})$ along the trajectory. When these errors are large relative to the underlying advantage signal (notably in our setting where correctness reward is largely terminal and intermediate signals are weak), the critic-induced noise can dominate and flip the sign of $\hat{A}_t$, leading to updates that penalize beneficial actions or reinforce suboptimal ones.

This issue is exacerbated by confidence-driven block decoding/remasking, where a single step updates many tokens and can induce abrupt deviations in the partial sequence state, making it substantially harder for a scalar value function to learn a coherent and stable baseline over the visited state manifold. We leave a systematic empirical characterization of such critic instability (e.g., correlation between critic-based advantages and return-based proxies) for future work.

# B. Experimental Details

### B.1. Training Details

We present the hyperparameters and experimental configurations utilized in our study. For the SDAR models, we employ a default block size of 32, dynamic decoding with a threshold of $\phi = 0.9$, top-$k = 0$, top-$p = 1$, and a temperature of 1.0. Regarding the rollout configuration, we utilize a block size of 32 with 32 denoising steps per block during the diffusion-based generation process. The maximum response length is set to 8192. In each training iteration, we sample 128 tasks and generate 32 responses per task. We use a temperature of 1.0 for exploration during training, while greedy decoding is applied for evaluation. Furthermore, we employ low-confidence dynamic remasking with a confidence threshold of 0.9 to selectively re-mask low-probability tokens during the denoising steps.

In terms of reward design, we adopt a formulation that incorporates both accuracy and speed signals. For mathematics tasks, we use binary outcomes as verifiable rewards based on answer equivalence checking, assigning a base reward of $\pm 1.0$ for correct or incorrect answers. We apply configurable filtering to retain only tasks where at least one response is correct and where the token-per-forward (TPF) variance exceeds a specific threshold (defaulting to 0.01). For coding tasks, we utilize the proportion of unit tests passed by the generated solutions as the reward. The speed reward is computed based on the TPF. Subsequently, we apply per-reward decoupled normalization.

The training procedure varies depending on the model components used. When a value model is incorporated, it is optimized using the mean squared error (MSE) loss on token-level returns. In such configurations, we select $(\gamma, \lambda) = (1, 1)$ and employ generalized advantage estimation (GAE) with $\lambda = 1.0$ for both pre-training and main training. The policy model employs a PPO-style clipped surrogate objective with $\epsilon = 0.2$. We utilize the KL divergence estimator $k = 3$ with a coefficient of $\beta = 0.01$, computed against a frozen reference model. Additionally, a negative log-likelihood (NLL) loss with a weight of 0.1 is applied to correct samples. The optimization process is performed using AdamW, with a learning rate of $1 \times 10^{-6}$ for the policy and $5 \times 10^{-6}$ for the value function (if applicable). We employ a constant learning rate scheduler with 10 warmup steps, a maximum gradient norm of 1.0, and gradient checkpointing. To accelerate the process, we implement distributed training across multiple GPU nodes; our experiments utilize configurations such as a single node equipped with 8 H200 GPUs, leveraging DeepSpeed ZeRO-1 optimization. Finally, for data processing, we employ sequential sampling with cursor-based traversal through the training dataset, which automatically reshuffles after the completion of a full pass.

## B.2. LightningRL Algorithm Pipeline

---

**Algorithm 1** LightningRL

---

1: **Input:**
2:  1) Training set $\mathcal{D} = \{q, a\}$ of prompts and reference answers
3:  2) Policy $\pi_\theta$; frozen reference policy $\pi_{\text{ref}}$
4:  3) Iterations $M$; target accepted prompt-groups per iteration $B$; rollouts per prompt $G$
5:  4) PPO clips $\epsilon$; update epochs $K$; optimizer lr $\eta$
6:  5) Reward/penalty weights: $\beta_{\text{KL}}, w_{\text{NLL}}$
7:  6) Filtering threshold $\tau_{\text{TPF}}$
8: **Initialize:** parameters $\theta$
9: **Initialize grouped dataset:** $\mathcal{D}_{\text{grp}} \leftarrow \emptyset$
10: **for** $t = 1$ to $M$ **do**
11:  **Freeze old policy:** $\pi_{\text{old}} \leftarrow \pi_\theta$   *// for ratios & KL*
12:  **Sample rollouts (dynamic sampling):**
13:  **while** $|\mathcal{D}_{\text{grp}}| < B$ **do**
14:   Sample one task $q \sim \mathcal{D}$
15:   **for** $j = 1$ to $G$ **do**
16:    Generate rollout $o^{(j)} \sim \pi_\theta(\cdot \mid q)$ using block-diffusion decoding
17:    Measure speed and correctness $r_{\text{tpf}}^{(j)} \leftarrow \text{TPF}^{(j)}, r_{\text{acc}}^{(j)} \leftarrow c^{(j)}$
18:   **end for**
19:   **Filter group:**
20:   **if** $\left( \sum_{j=1}^{G} c^{(j)} = 0 \right)$ **or** $\left( \max_j r_{\text{tpf}}^{(j)} - \min_j r_{\text{tpf}}^{(j)} < \tau_{\text{TPF}} \right)$ **then**
21:    Reject group    *// all-wrong group and near-tied speed signal*
22:    **continue**
23:   **end if**
24:   **Decoupled normalization within group:**
25:    Group stats: $\mu_{\text{acc}} \leftarrow \text{mean}_j(r_{\text{acc}}^{(j)}), \ \sigma_{\text{acc}} \leftarrow \text{std}_j(r_{\text{acc}}^{(j)})$
26:    $\mu_{\text{tpf}} \leftarrow \text{mean}_j(r_{\text{tpf}}^{(j)}), \ \sigma_{\text{tpf}} \leftarrow \text{std}_j(r_{\text{tpf}}^{(j)})$
27:    Group-relative advantages $A_{\text{acc}}^{(j)} \leftarrow \dfrac{r_{\text{acc}}^{(j)} - \mu_{\text{acc}}}{\sigma_{\text{acc}} + \varepsilon}, A_{\text{tpf}}^{(j)} \leftarrow \dfrac{r_{\text{tpf}}^{(j)} - \mu_{\text{tpf}}}{\sigma_{\text{tpf}} + \varepsilon}$   *// per-reward decoupled normalization*
28:    Aggregate objectives $A^{(i,j)} \leftarrow A_{\text{acc}}^{(i,j)} + A_{\text{tpf}}^{(i,j)}$
29:    Group normalization $\hat{A}^{(i,j)} \leftarrow \dfrac{A^{(i,j)} - \bar{A}}{\sigma_A + \varepsilon}$
30:    Store accepted group $\mathcal{D}_{\text{grp}} \leftarrow (q, \{o^{(j)}\}_{j=1}^{G}, \{\hat{A}^{(j)}\}, \{c^{(j)}\})$
31:  **end while**
32:  **Policy optimization:**
33:  **for** $e = 1$ to $K$ **do**
34:   Sample minibatch $\mathcal{M} \subset \mathcal{D}_{\text{grp}}$ ; Compute ratios $\rho^{(i,j)} \leftarrow \frac{\pi_\theta(o^{(i,j)}|q^{(i)})}{\pi_{\text{old}}(o^{(i,j)}|q^{(i)})}$
35:   **Compute losses:** $\mathcal{L} \leftarrow \mathcal{L}_{\text{PG}}(\rho, A) + \beta_{\text{KL}}\mathcal{L}_{\text{KL}}(\pi_\theta, \pi_{\text{ref}}) + w_{\text{NLL}}\mathcal{L}_{\text{NLL}}(\pi_\theta; c)$
36:   **Update:** $\theta \leftarrow \theta - \eta \nabla_\theta \mathcal{L}$
37:  **end for**
38: **end for**
39: **Output:** trained policy $\pi_\theta$

---

# C. Additional Experiments

## C.1. Scalability and Efficiency Analysis

As summarized in Tab. 6, we evaluate scalability along two primary axes. First, regarding model scale: under a fixed block size of 32, scaling from 1.7B to 8B parameters significantly boosts both reasoning accuracy and parallelism. Notably, while the baseline's throughput remains stagnant as model size grows, LightningRL effectively leverages increased capacity to unlock higher parallelism. Second, regarding block size: increasing the block size on the 8B model raises the upper bound for parallel decoding, an effect especially prominent on the MBPP benchmark. Overall, LightningRL scales favorably and remains robustly effective across different model sizes and decoding configurations.

*Table 6.* **Comparison of SDAR and LightningRL under identical settings.** The results are grouped by Model Scale and Block Size (BS) to facilitate direct comparison across four datasets. The AUP in this table is computed using each model's own accuracy at TPF≈1 as $y_{max}$.

| Scale | BS | Model | GSM8K | | | MATH500 | | | MBPP | | | HumanEval | | |
|---|---|---|---|---|---|---|---|---|---|---|---|---|---|---|
| | | | Acc (%) | TPF | AUP | Acc (%) | TPF | AUP | Acc (%) | TPF | AUP | Acc (%) | TPF | AUP |
| 1.7B | 32 | SDAR | 71.5 | 2.48 | 176.4 | 41.2 | 5.62 | 226.3 | 39.0 | 3.86 | 149.8 | 48.8 | 2.81 | 136.0 |
| | | **LightningRL** | 71.7 | **3.40** | **251.4** | 41.2 | **6.01** | **241.2** | 37.3 | **5.07** | **185.8** | 48.2 | **3.43** | **165.2** |
| 4B | 32 | SDAR | 86.6 | 3.10 | 243.8 | 53.6 | 5.09 | 263.9 | 54.0 | 1.96 | 105.9 | 59.8 | 3.67 | 216.8 |
| | | **LightningRL** | 85.4 | **4.37** | **374.7** | 56.4 | **6.55** | **358.7** | 52.2 | **3.05** | **158.3** | 57.3 | **4.35** | **242.3** |
| 8B | 32 | SDAR | 88.9 | 2.85 | 252.5 | 63.6 | 4.81 | 299.5 | 58.0 | 2.44 | 81.1 | 73.5 | 2.39 | 123.8 |
| | | **LightningRL** | 90.3 | **5.58** | **492.4** | 63.0 | **6.28** | **407.5** | 58.3 | **11.10** | **641.6** | 72.6 | **6.30** | **450.1** |
| | 8 | SDAR | 91.0 | 2.96 | 269.1 | 64.1 | 3.53 | 224.8 | 59.6 | 1.72 | 103.3 | 76.9 | 2.77 | 211.6 |
| | | **LightningRL** | 89.4 | **3.75** | **331.8** | 67.0 | **4.21** | **279.5** | 58.2 | **3.11** | **179.3** | 76.8 | **3.30** | **258.8** |
| | 4 | SDAR | 91.1 | 2.35 | 213.9 | 71.2 | 2.49 | 176.8 | 63.3 | 1.84 | 116.7 | 78.6 | 1.53 | 120.3 |
| | | **LightningRL** | 91.0 | **3.21** | **291.6** | 70.3 | **3.42** | **237.7** | 63.2 | **2.47** | **155.5** | 78.2 | **2.32** | **181.3** |

## C.2. Wall-Clock Speed Comparison

*Table 7.* **Tokens Per Second (TPS) performance comparison on H100 GPUs.** We report on GSM8K datasets. LightningRL achieves superior throughput.

| Model | H100 TPS | TPF | Acc (%) |
|---|---|---|---|
| Qwen-2.5-7B-it | 57.3 | 1.00 | 74.1 |
| Fast-dLLM-v2 | 150.0 | 2.21 | 77.5 |
| dParallel-LLaDA | 172.2 | 5.14 | 72.6 |
| d3LLM-LLaDA | 288.9 | **9.11** | 73.1 |
| SDAR | 105.6 | 2.85 | 88.9 |
| **LightningRL** | **336.0** | 5.58 | **90.3** |

We conduct TPS benchmarks for SDAR using the SGLang on single-device H100 configurations (Tensor Parallel Size 1), with results summarized in Table 7. Since prior works are not yet natively supported by SGLang, their reported results from the original publication are cited for comparison (Qian et al., 2026). Notably, LightningRL achieves exceptional inference speed, significantly outperforming the baselines.

## C.3. Training Dynamics Details

We plot the training curves of LightningRL on GSM8K in Fig. 8a-8c. For comparison, we apply TraceRL (Wang et al., 2025b), a commonly used RL framework for dLLMs, to the same training settings (including rewards, hyperparameters, etc.). As shown, during the training of TraceRL, the optimization for speed progressively erodes the correctness signal, and the accuracy reward drops sharply while speed gains remain limited. The training ultimately collapses. In contrast, LightningRL converges stably and avoids this collapse behavior, maintaining the accuracy signal while improving speed, resulting in a consistently better efficiency–accuracy frontier.

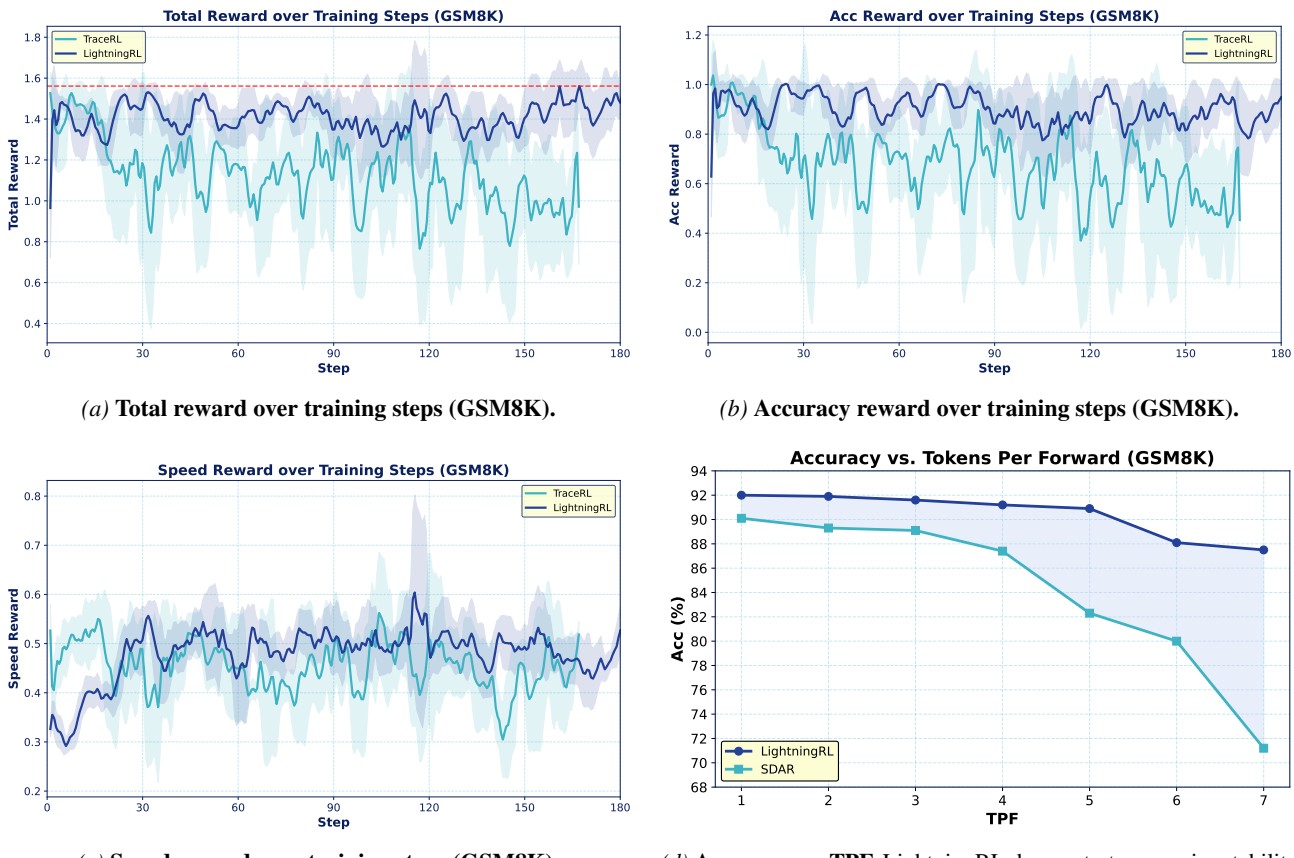

*(a)* **Total reward over training steps (GSM8K).**

*(b)* **Accuracy reward over training steps (GSM8K).**

*(c)* **Speed reward over training steps (GSM8K).**

*(d)* **Accuracy vs. TPF.** LightningRL demonstrates superior stability at higher Tokens Per Forward settings, significantly widening the accuracy gap over SDAR, which drops sharply after TPF 4.

*Figure 8.* **Training dynamics and Accuracy-Parallelism Trade-off on GSM8K.** LightningRL avoids the objective drift and reward collapse observed in TraceRL, while sustaining higher decoding throughput and more synchronized termination.

## C.4. Accuracy–Parallelism Trade-off

We further investigate the inherent trade-off between decoding parallelism and reasoning accuracy on the GSM8K benchmark. In non-autoregressive or drafting-based decoding frameworks, increasing the number of Tokens Per Forward (TPF) naturally improves throughput and parallelism. However, this aggressively challenges the model's reasoning capabilities, as it must predict longer sequences without the benefit of step-by-step autoregressive feedback, typically leading to a degradation in accuracy. As illustrated in Fig. 8d, while both methods experience accuracy degradation as TPF increases from 1 to 7, their trajectories differ significantly. The baseline SDAR is highly vulnerable to larger TPF settings; its accuracy drops precipitously after $TPF = 3$, falling from $87.4\%$ at $TPF = 4$ to just $71.2\%$ at $TPF = 7$. In contrast, LightningRL demonstrates remarkable robustness against this accuracy-parallelism trade-off. Its performance degrades only marginally, maintaining a high accuracy of $87.5\%$ even at the aggressive setting of $TPF = 7$. These results confirm that LightningRL significantly pushes the Pareto frontier, mitigating the performance penalty of larger forward steps to enable higher decoding throughput without sacrificing reasoning quality.

## C.5. Hyperparameter Sensitivity

We conduct a sensitivity study on key hyperparameters of LightningRL. Unless otherwise noted, all experiments use the 8B model with a block size of 32 on GSM8K.

*Table 8.* Sensitivity to $\mu$ ($\delta = 0.01$ fixed).

| $\mu$ | 0.0 | 0.1 (ours) | 0.2 | 0.4 | 0.8 | 1.0 |
|---|---|---|---|---|---|---|
| Acc (%) | 84.7 | **90.3** | 88.4 | 86.7 | 86.3 | 87.1 |
| TPF | 4.43 | **5.58** | 5.66 | 5.72 | 5.26 | 5.21 |
| AUP | 385.7 | **507.5** | 491.0 | 448.6 | 443.3 | 441.7 |

*Table 9.* Sensitivity to $\delta$ ($\mu = 0.1$ fixed).

| $\delta$ | 0.0 | 0.01 (ours) | 0.05 | 0.1 |
|---|---|---|---|---|
| Acc (%) | 84.1 | **90.3** | 88.3 | 88.6 |
| TPF | 4.96 | **5.58** | 5.32 | 5.70 |
| AUP | 402.7 | **507.5** | 461.3 | 488.1 |

Tab. 8 reports results for varying $\mu$ with $\delta = 0.01$ fixed. Removing the NLL anchoring term entirely ($\mu = 0$) causes a substantial drop in both accuracy (84.7%) and AUP (385.7), confirming that the NLL loss plays a critical role in stabilizing the accuracy objective. As $\mu$ increases beyond 0.1, accuracy gradually declines because an excessively strong NLL term dominates the policy gradient and restricts exploration. The selected value $\mu = 0.1$ achieves the highest AUP (507.5) by striking a balance between anchoring correctness and retaining sufficient policy flexibility.

Tab. 9 reports results for varying $\delta$ with $\mu = 0.1$ fixed. When filtering is disabled ($\delta = 0$), both accuracy and AUP degrade, as near-tied groups dilute the effective learning signal. Larger thresholds ($\delta = 0.05$ and 0.1) discard more groups and preserve reasonable accuracy, yet the reduced sample diversity leads to slightly lower AUP compared to $\delta = 0.01$.

We additionally examine sampling temperature and group size, as both directly influence the informativeness of group-relative RL signals.

*Table 10.* Sensitivity to sampling temperature.

| Temperature | 0.2 | 0.4 | 0.6 | 0.8 | 1.0 |
|---|---|---|---|---|---|
| Acc (%) | 28.4 | 81.3 | 89.1 | 90.5 | **92.9** |

*Table 11.* Sensitivity to group size.

| Group size | 4 | 8 | 16 | 32 | 64 |
|---|---|---|---|---|---|
| Nonzero adv. | 1.08M | 2.18M | 4.63M | 9.41M | 17.81M |
| Collapse Ratio | 0.0137 | 0.0111 | 0.0114 | 0.0122 | 0.0144 |

As shown in Tab. 10, rollout accuracy is highly sensitive to the sampling temperature. Performance improves substantially as the temperature increases within the tested range, with the best result achieved at 1.0. A higher temperature not only produces the highest rollout quality in our experiments but also better preserves exploration, which is essential for generating diverse candidates and yielding informative within-group comparisons during RL training.

Tab. 11 shows the effect of group size. As the group size increases, the number of nonzero advantages grows steadily, indicating that more sampled trajectories contribute meaningful learning signals. Meanwhile, the collapse ratio first decreases and then rises only mildly. These results suggest that a moderately large group size enriches the effective advantage signal without inducing severe collapse. However, very large group sizes incur substantially higher memory and computational costs, so we adopt $G = 32$ as a practical trade-off among signal richness, training stability, and efficiency.

In summary, LightningRL exhibits robust performance across a moderate range of all four hyperparameters, rather than depending on a single sensitive configuration. The default setting ($\mu = 0.1$, $\delta = 0.01$, temperature $= 1.0$, $G = 32$) achieves the strongest overall accuracy-parallelism trade-off among all tested values.

