# OpenReview forum: "LightningRL: Breaking the Accuracy–Parallelism Trade-off of Block-wise dLLMs via Reinforcement Learning"
_ICML.cc/2026/Conference — ICML 2026 regular_

### Official Review · Reviewer_TXxQ · 2026-02-28

**Soundness:** 2
**Presentation:** 2
**Significance:** 2
**Originality:** 3
**Overall Recommendation:** 3
**Confidence:** 3

**Summary:**

This paper proposes the LightningRL framework based on reinforced learning, which breaks the accuracy-parallelism trade-off of block-wise dLLMs via three key innovations, achieves superior speed-quality performance on math and code tasks, and provides a practical post-training optimization solution for high-throughput dLLM deployment with strong scalability.

**Compliance With Llm Reviewing Policy:**

Affirmed.

**Final Justification:**

Final Review Comment (Maintaining Original Rejection Recommendation)

Thank the authors for their detailed response and supplementary experiments addressing the initial review comments. While the added state-of-the-art comparisons and hyperparameter sensitivity analyses partially address my concerns about insufficient baseline comparison and inadequate hyperparameter discussion, critical flaws remain in the paper: the ablation studies are still incomplete, and the effectiveness of core components (e.g., token-level NLL regularization) is not sufficiently validated, failing to fully support the reliability and novelty of the proposed method. Additionally, the shortcomings in experimental completeness and method reproducibility have not been thoroughly resolved. Therefore, I maintain my original "Reject" recommendation, and suggest the authors resubmit after further improving the experimental design and supplementing comprehensive ablation validation for core components.

Dimensional Trade-off Summary (Concise Version)

Originality: The method introduces RL-based joint optimization for the accuracy-parallelism trade-off in dLM training, which has certain novelty, but the effectiveness of the core innovation is not sufficiently validated through comprehensive ablation studies, weakening the persuasiveness of its originality.

Significance: The research direction is valuable, but experimental flaws undermine the practical value of the method, making it impossible to fully demonstrate its advantages in dLM deployment.

Soundness: The supplementary experiments partially mitigate the issue of insufficient comparison, but the validation of core
components and robustness analysis of hyperparameters remain inadequate, failing to meet the soundness standard.

Clarity: The authors have supplemented some content, but there is still room for improvement in the overall experimental completeness and logical rigor of the paper.

The authors' rebuttal has not fully resolved my core concerns regarding the paper's experimental credibility and the effectiveness of core components, so I maintain my original evaluation score and rejection recommendation.

**Key Questions For Authors:**

see weakness

**Limitations:**

Insufficient experiments and incomplete ablation studies compromise the work’s credibility, with no adequate validation for core components and their hyperparameter impacts.

**Strengths And Weaknesses:**

- Weaknesses
1. The comparison set is insufficient, failing to include critical state-of-the-art methods like DirL and Co-GRPO, which undermines the persuasiveness of the proposed approach’s competitive advantages.

2. There is insufficient discussion on hyperparameters such as group size, temperature coefficient, and loss weight.

3. The effectiveness of per-reward decoupled normalization needs to be further verified, as accuracy and TPF are not completely independent of each other. Additionally, the token-level NLL regularization faces the risk of overfitting, and the ablation experiments for this component are insufficient.

- Strengths

1. Existing methods mostly focus only on a single goal of speed or accuracy, while this paper treats parallelism as a first-class training goal and jointly optimizes both through reinforcement learning, filling a key gap in dLLM acceleration research.

---

> ### Author Rebuttal · Authors · 2026-03-31
>
> We thank the reviewer for the careful reading, constructive feedback, and time spent on evaluating our work. Below, we address each concern in turn.
>
> ---
> **W1: Comparison with State-of-the-art Methods.**
> >The comparison set is insufficient, failing to include critical state-of-the-art methods like DirL and Co-GRPO, which undermines the persuasiveness of the proposed approach’s competitive advantages.
>
> We agree that direct comparison with recent RL-based dLLM post-training methods would strengthen the paper. We have therefore completed additional experiments against Couple-GRPO, TraceRL, and DiRL under the same evaluation setup, and summarize the results below:
> |         Method         |  GSM8K (Acc/TPF/AUP)  | MATH500 (Acc/TPF/AUP) |   MBPP (Acc/TPF/AUP)    | HumanEval (Acc/TPF/AUP) |
> | :--------------------: | :-------------------: | :-------------------: | :---------------------: | :---------------------: |
> |      Couple-GRPO       |   75.3/ 4.22/ 254.7   |   59.1/ 4.93/ 283.1   |    54.0/ 2.61/ 139.9    |    68.2/ 2.42/ 164.6    |
> |        TraceRL         |   76.9/ 5.04/ 378.6   |   60.6/ 4.82/ 284.6   |    57.8/ 2.50/ 144.2    |    75.0/ 2.29/ 171.6    |
> |          DiRL          |   86.6/ 4.87/ 414.1   |   61.6/ 5.04/ 301.2   |    56.8/ 2.70/ 152.7    |    76.0/ 2.33/ 176.7    |
> | **LightningRL (Ours)** | **90.3/ 5.58/ 492.4** | **63.0/ 6.28/ 407.5** | **58.3/ 11.10 / 641.6** |  **72.6/ 6.30/ 450.1**  |
>
> These additional results further strengthen our main claim that LightningRL achieves the strongest overall accuracy–parallelism trade-off among RL-based dLLM post-training methods. The advantage is consistent with our method design for this multi-objective RL: decoupled normalization preserves informative comparisons, the token-level NLL anchor stabilizes correctness, and TPF-aware dynamic sampling improves optimization efficiency. Together, these components lead to a better accuracy–parallelism trade-off.
>
> ---
> **W2: Hyperparameters.**
> >There is insufficient discussion on hyperparameters such as group size, temperature coefficient, and loss weight.
>
> We agree that the hyperparameter discussion can be strengthened. For group size and sampling temperature, we follow the commonly used rollout settings in prior group-wise RL methods [1]. We will include a concise sensitivity study on both in the final version. Beyond these shared rollout settings, the main new hyperparameters introduced by LightningRL are the NLL loss weight $\mu$ and the TPF-aware filtering threshold $\delta$. We therefore conducted additional sensitivity analyses on both, summarized below.
>
> | NLL weight $\mu$ |  0.0  | 0.1(ours) |  0.2  |  0.4  |  0.8  |  1.0  |
> | :--------------: | :---: | :-------: | :---: | :---: | :---: | :---: |
> |     Acc (%)      | 84.7  | **90.3**  | 88.4  | 86.7  | 86.3  | 87.1  |
> |       TPF        | 4.43  | **5.58**  | 5.66  | 5.72  | 5.26  | 5.21  |
> |       AUP        | 385.7 | **492.4** | 491.0 | 448.6 | 443.3 | 441.7 |
>
> | TPF threshold $\delta$ |  0.0  | 0.01(ours) | 0.05  |  0.1  |
> | :--------------------: | :---: | :--------: | :---: | :---: |
> |        Acc (%)         | 84.1  |  **90.3**  | 88.3  | 88.6  |
> |          TPF           | 4.96  |  **5.58**  | 5.32  | 5.70  |
> |          AUP           | 402.7 | **492.4**  | 461.3 | 488.1 |
>
> These results support our default choices of a moderate NLL weight and a small but non-zero filtering threshold. We will include these sensitivity results in the appendix.
>
> ---
> **W3 Decoupled Normalization and NLL.**
> >The effectiveness of per-reward decoupled normalization needs to be further verified, as accuracy and TPF are not completely independent of each other. Additionally, the token-level NLL regularization faces the risk of overfitting, and the ablation experiments for this component are insufficient.
>
> We have verified the effect of decoupled normalization empirically. As shown in Fig. 3 and Tab. 3, it reduces collapse, improves training stability, and its removal degrades performance from 90.3 / 5.58 to 83.2 / 4.79 (Acc / TPF). Besides, we clarify that our per-reward decoupled normalization does not assume that accuracy and TPF are fully independent. Rather, it is designed to handle their different scales and granularities, since directly combining a coarse accuracy reward with a finer-grained TPF reward can create within-group ties or near-ties and weaken the training signal. This is also consistent with the motivation of concurrent work [2].
>
> For the token-level NLL term, we agree that overfitting is a risk when its weight is too large. In our additional $\mu$-ablation (W2), we observe that when $\mu > 0.4$, the training has not yet converged, but evaluation accuracy already declines, while the optimization increasingly drifts toward the parallelism reward. This suggests that the NLL term should act as a light anchor rather than a dominant objective.
>
> [1] Wang et al. “TraceRL” ArXiv: 2509.06949
>
> [2] Liu et al. “GDPO” ArXiv: 2601.05242

---

> > ### Author Rebuttal · Reviewer_TXxQ · 2026-04-02
> >
> > Thank you for your response and efforts to address our concerns. We have reviewed the supplementary experiments and clarifications. While progress has been made, the validation of core components and completeness of SOTA comparison still require further strengthening. We will adjust our recommendation to Weak Reject / Borderline pending the final revised manuscript.

---

> > > ### Author Response · Authors · 2026-04-05
> > >
> > > We thank the reviewer again for the careful follow-up and constructive feedback. We appreciate that progress has been made, and we will continue to further strengthen our empirical validation and clarify the scope of our comparisons.
> > >
> > > ---
> > > **Hyperparameters**
> > >
> > > We further analyzed both sampling temperature and group size, since they directly affect the informativeness of group-relative RL signals.
> > >
> > > For **temperature**, we observe that rollout accuracy is highly sensitive to the sampling temperature.On GSM8K, performance improves substantially as the temperature increases within the tested range, with the best result achieved at 1.0:
> > > | Temperature |  0.2  |  0.4  |  0.6  |  0.8  |  1.0  |
> > > | :---------: | :---: | :---: | :---: | :---: | :---: |
> > > |   Acc(%)    | 28.4% | 81.3% | 89.1% | 90.5% | 92.9% |
> > >
> > > These results support our use of temperature = 1.0. A higher temperature not only gives the best rollout quality in our experiments, but also better preserves exploration, which is important for generating diverse candidates and producing informative within-group comparisons during RL training.
> > >
> > > For **group size**, we observe that increasing the group size steadily increases the number of non-zero advantages, indicating that more sampled trajectories contribute meaningful learning signals. At the same time, the collapse ratio first decreases and then increases only mildly:
> > > |           Group size           |    4    |    8    |   16    |   32    |    64    |
> > > | :----------------------------: | :-----: | :-----: | :-----: | :-----: | :------: |
> > > |     Num nonzero advantages     | 1083512 | 2181156 | 4632409 | 9406101 | 17810059 |
> > > | Collapse Ratios( $\tau=0.1$) | 0.0137  | 0.0111  | 0.0114  | 0.0122  |  0.0144  |
> > >
> > > Overall, these results suggest that using a moderately large group size is beneficial: it enriches effective advantage signals without causing severe collapse. Meanwhile, very large group sizes impose substantially higher memory and computational costs. This is why we use G = 32 as a practical trade-off between signal richness, stability, and training efficiency.
> > >
> > > ---
> > > **Validation of Decoupled Normalization**
> > >
> > > We further validate decoupled normalization at the advantage-signal level, rather than only through final Acc/TPF (as shown in Fig. 3 and the Tab. 3). As shown in the table below, the full normalization pipeline increases the number of nonzero advantages (**691,440 → 716,908**), reduces the collapse ratio (**0.062 → 0.036**), and lowers the standard deviation of advantages (**1.245 → 1.062**). In contrast, applying normalization to only one reward yields only limited gains.
> > > |              Case              | Naive  | +TPF Normalization | + Acc Normalization | + Batch Normalization |
> > > | :----------------------------: | :----: | :----------------: | :-----------------: | :-------------------: |
> > > |     Num nonzero advantages     | 691440 |       703178       |       684771        |        716908         |
> > > | Collapse Ratios( $\tau=0.1$) | 0.062  |       0.059        |        0.059        |         0.036         |
> > > |         Std advantage          | 1.245  |       1.132        |        1.153        |         1.062         |
> > >
> > > This supports our claim that decoupled normalization stabilizes group-relative credit assignment under mixed-granularity rewards: correctness is coarse/discrete, while TPF is finer-grained/continuous, so naive joint normalization can weaken the correctness signal through degenerate within-group comparisons.
> > >
> > > ---
> > > **SOTA Comparison**
> > >
> > > In our earlier rebuttal, we already added more detailed comparisons against strong baselines in dLLMs. Regarding the reviewer’s suggestion to include **Co-GRPO**, we would like to clarify that, although it is also an RL method for MDMs, it is proposed for **image generation** rather than **language generation**. For this reason, we did not include it in the empirical comparison as a dLLM SOTA baseline.
> > > That said, we agree that the revised manuscript should make the baseline selection criteria more explicit, so that the scope of the comparison is easier to interpret. We will revise this part accordingly and further clarify the relationship between our method and adjacent RL-based diffusion work outside the language setting.
> > > We thank the reviewer again for the constructive feedback. We will continue strengthening the final manuscript by making these analyses and comparison criteria clearer, and by further improving the empirical validation of our method. If there are additional experiments or analyses that the reviewer would find particularly important, we would be happy to prioritize them and include them in the revision.

---

### Official Review · Reviewer_b5jb · 2026-03-07

**Soundness:** 3
**Presentation:** 3
**Significance:** 3
**Originality:** 3
**Overall Recommendation:** 4
**Confidence:** 2

**Summary:**

This paper addresses the accuracy-parallelism trade-off in block-wise diffusion language models (dLLMs), where increasing tokens per forward pass (TPF) through aggressive parallel decoding typically degrades task accuracy. The authors propose LightningRL, a reinforcement learning-based post-training approach built on the GRPO framework, to directly optimize the speed-quality Pareto frontier.

**Compliance With Llm Reviewing Policy:**

Affirmed.

**Key Questions For Authors:**

What is the actual wall-clock inference speedup of LightningRL compared to SDAR and other baselines on the same hardware? TPF alone does not capture the full picture of practical efficiency gains. Concrete latency numbers would significantly strengthen the practical impact claim.

Can you report the training cost (GPU hours, total rollouts, training tokens processed) for LightningRL post-training? This is crucial for understanding the practical feasibility of this approach.

Have you tested LightningRL on other block-wise dLLM architectures beyond SDAR? If yes, do the improvements transfer? If not, what prevents this generalization?

**Limitations:**

The authors do not extensively discuss limitations. Key missing discussions include: (1) the method's dependence on a single base architecture (SDAR), which limits generalizability claims; (2) the lack of wall-clock inference time measurements; (3) the RL training cost, which could be substantial given the need for multiple rollouts per prompt.

**Strengths And Weaknesses:**

Strengths:

The accuracy-parallelism trade-off is a genuine bottleneck for practical deployment of dLLMs. Framing this as an RL problem is a natural and effective formulation. Table 3 clearly shows each component's contribution, and the training dynamics comparison with TraceRL (Fig. 6) provides valuable insight into why prior RL approaches fail (reward collapse). The paper systematically evaluates across model scales (1.7B, 4B, 8B) and block sizes (4, 8, 32), demonstrating consistent improvements.

Weaknesses:

The paper compares primarily against TraceRL for the RL training dynamics comparison. Other RL frameworks for dLLMs (e.g., those mentioned in Related Work Section 5.2) are not empirically compared in the main experiments.

While LightningRL improves TPF significantly, at block size 32 the accuracy on some benchmarks (e.g., HumanEval: 72.6% vs. SDAR's 74.4% at b=32) actually drops slightly. The paper should discuss when accuracy trade-offs become unacceptable.

---

> ### Author Rebuttal · Authors · 2026-03-31
>
> We thank the reviewer for the careful reading and thoughtful feedback. We appreciate the positive assessment of our work, and below we address the concerns raised.
>
> ---
> **W1: Comparison with Recent RL Methods.**
> >The paper compares primarily against TraceRL for the RL training dynamics comparison. Other RL frameworks for dLLMs (e.g., those mentioned in Related Work Section 5.2) are not empirically compared in the main experiments.
>
> We have now completed additional empirical comparisons with recent RL methods, including Couple-GRPO, TraceRL, and DiRL. The results are summarized below:
>
> |         Method         |  GSM8K (Acc/TPF/AUP)  | MATH500 (Acc/TPF/AUP) |   MBPP (Acc/TPF/AUP)    | HumanEval (Acc/TPF/AUP) |
> | :--------------------: | :-------------------: | :-------------------: | :---------------------: | :---------------------: |
> |      Couple-GRPO       |   75.3/ 4.22/ 254.7   |   59.1/ 4.93/ 283.1   |    54.0/ 2.61/ 139.9    |    68.2/ 2.42/ 164.6    |
> |        TraceRL         |   76.9/ 5.04/ 378.6   |   60.6/ 4.82/ 284.6   |    57.8/ 2.50/ 144.2    |    75.0/ 2.29/ 171.6    |
> |          DiRL          |   86.6/ 4.87/ 414.1   |   61.6/ 5.04/ 301.2   |    56.8/ 2.70/ 152.7    |    76.0/ 2.33/ 176.7    |
> | **LightningRL (Ours)** | **90.3/ 5.58/ 492.4** | **63.0/ 6.28/ 407.5** | **58.3/ 11.10 / 641.6** |  **72.6/ 6.30/ 450.1**  |
>
> These results further support our main claim: LightningRL achieves a substantially better accuracy-parallelism frontier, with the strongest overall AUP and markedly higher parallelism across benchmarks.
>
> ---
> **W2: Trade-off.**
> >The reviewer notes that, although LightningRL substantially improves TPF, it also leads to slight accuracy drops on some benchmarks at block size 32 (e.g., HumanEval: 72.6% vs. SDAR’s 74.4%). The reviewer therefore asks the paper to clarify when such accuracy trade-offs become unacceptable.
>
> For applications that require near-lossless accuracy, an intermediate checkpoint already provides a strong trade-off: on HumanEval, it achieves **74.2%** accuracy at **6.03** TPF, closely matching the SDAR baseline (**74.4%**) while still delivering a substantial **2.5×** gain in parallelism (**2.39 → 6.03**).
>
> In our study, we treat an accuracy drop of **within 2%** as an acceptable trade-off, since our goal is to maximize parallelism while preserving near-lossless accuracy. We will clarify this practical criterion in the revised manuscript.
>
> ---
> **Q1: Wall-clock Inference Speedup.**
> >What is the actual wall-clock inference speedup of LightningRL compared to SDAR and other baselines on the same hardware? TPF alone does not capture the full picture of practical efficiency gains.
>
> We further report wall-clock TPS measurements for LightningRL and the SDAR baseline on GSM8K, using SGLang on single-device H100 configurations, with results summarized below:
> |         Method         | Accuracy (%) | Speed (TPS) |
> | :--------------------: | :----------: | :---------: |
> |     Qwen-2.5-7B-it     |     57.3     |    74.1     |
> |      Fast-dLLM-v2      |     77.5     |    150.0    |
> |    dParallel-LLaDA     |     72.6     |    172.2    |
> |      d3LLM-LLaDA       |     73.1     |    288.9    |
> |    SDAR (Baseline)     |     88.9     |    105.6    |
> | **LightningRL (Ours)** |   **90.3**   |  **336.0**  |
>
> Since several prior dLLM baselines are not yet natively supported in our SGLang setup, we cite their throughput results from the original publication for reference [1]. Notably, these results are also measured on GSM8K with single-device H100, making the comparison directly aligned.LightningRL demonstrates substantially stronger practical inference efficiency and delivers the highest measured throughput among the compared methods.
>
> [1] Qian et al. “d3LLM.” ArXiv: 2601.07568
>
> ---
> **Q2: Training Cost.**
> >Can you report the training cost (GPU hours, total rollouts, training tokens processed) for LightningRL post-training? This is crucial for understanding the practical feasibility of this approach.
>
> Yes. For the GSM8K dataset train split, the post-training cost of LightningRL is **88.5** GPU hours, **86,016** total rollouts, and **21** epochs, corresponding to approximately **41.04M** tokens. The training costs on the other datasets are also close, with only minor variation across datasets and configurations. We will add these statistics to the revised manuscript.
>
> ---
> **Q3: Generalization.**
> >Have you tested LightningRL on other block-wise dLLM architectures beyond SDAR? If yes, do the improvements transfer? If not, what prevents this generalization?
>
> Yes. We have successfully generalized LightningRL beyond SDAR to Fast-dLLM-v2. On GSM8K, post-training improves Fast-dLLM-v2 from **77.5% / 2.21 TPF to 78.6% / 4.12 TPF**, showing that the method remains effective on another block-wise dLLM architecture. In addition, our results across multiple model scales and block sizes already indicate that the final improvement depends on the underlying backbone and decoding configuration.

---

> > ### Author Rebuttal · Reviewer_b5jb · 2026-04-02
> >
> > Thanks. My major concerns have been explained. I will keep my positive score.

---

> > > ### Author Response · Authors · 2026-04-05
> > >
> > > Thank you for taking the time to read our rebuttal. We are very glad to hear that our explanations have successfully addressed your major concerns. We deeply appreciate your constructive feedback during the review process and your continued support for our work.

---

### Official Review · Reviewer_LDYx · 2026-03-17

**Soundness:** 2
**Presentation:** 2
**Significance:** 2
**Originality:** 2
**Overall Recommendation:** 4
**Confidence:** 3

**Summary:**

For block-wise diffusion LLMs, increasing tokens-per-forward (TPF) via aggressive confidence-threshold decoding consistently degrades task accuracy. Prior work have designed methods for this through training-free sampling heuristics or distillation methods. In this work, the authors propose to use post-training RL that builds on GRPO with an objective that optimizes a joint reward over final accuracy and overall TPF.

They trained LightningRL-8B-b32, which is tuned from SDAR-8B, and achieves higher TPF than baseline and higher AUP.

**Compliance With Llm Reviewing Policy:**

Affirmed.

**Final Justification:**

my questions have been adequately addressed. i remain my positive score.

**Key Questions For Authors:**

please see above

**Limitations:**

please see above

**Strengths And Weaknesses:**

pros:

1. The paper propose an insightful method where the authors identified the training-free and distillation-based approaches for accuracy-parallelism trade-off in block-wise dLLMs are insufficient. post-training RL is a new angle for solving this.
2. The robustness and scalability of their proposed methods holds across model sizes and block sizes (4, 8, 32), with larger block sizes particularly unlocking higher parallelism on code tasks.

cons:
1. lack of sensitivity analysis or ablation on crucial hyperparameters, such as the TPF-aware filtering threshold.
2. The paper uses a joint reward over accuracy and TPF but has not provided the explicit reward formulation

---

> ### Author Rebuttal · Authors · 2026-03-31
>
> We thank the reviewer for identifying post-training RL as a valuable new perspective for optimizing dLLMs and for acknowledging the robustness and scalability of our method across different model sizes and block configurations.
>
> ---
> **Cons1: Hyperparameters.**
> >lack of sensitivity analysis or ablation on crucial hyperparameters, such as the TPF-aware filtering threshold.
>
> We thank the reviewer for this helpful suggestion. In response, we conducted an additional sensitivity study on the token-level NLL weight $\mu$and the TPF-aware filtering threshold $\delta$, and summarize the results below.
> | NLL weight $\mu$ |  0.0  | 0.1(ours) |  0.2  |  0.4  |  0.8  |  1.0  |
> | :--------------: | :---: | :-------: | :---: | :---: | :---: | :---: |
> |     Acc (%)      | 84.7  | **90.3**  | 88.4  | 86.7  | 86.3  | 87.1  |
> |       TPF        | 4.43  | **5.58**  | 5.66  | 5.72  | 5.26  | 5.21  |
> |       AUP        | 385.7 | **492.4** | 491.0 | 448.6 | 443.3 | 441.7 |
>
> | TPF threshold $\delta$ |  0.0  | 0.01(ours) | 0.05  |  0.1  |
> | :--------------------: | :---: | :--------: | :---: | :---: |
> |        Acc (%)         | 84.1  |  **90.3**  | 88.3  | 88.6  |
> |          TPF           | 4.96  |  **5.58**  | 5.32  | 5.70  |
> |          AUP           | 402.7 | **492.4**  | 461.3 | 488.1 |
>
> These results show that LightningRL is reasonably robust across a moderate range of $\mu$ and $\delta$, rather than relying on a single brittle setting. At the same time, our final choice $(\mu=0.1,\delta=0.01)$ gives the strongest overall accuracy–parallelism trade-off among the tested values, especially in terms of AUP.
>
> LightningRL optimizes a weighted sum of the policy, KL, and NLL losses, where a reduction strategy defines how token-wise losses from sampled rollouts are aggregated into the final scalar objective. We compare token-level averaging, which upweights longer rollouts, with sequence-level averaging, which gives each rollout equal weight. Since rollout lengths vary, the reduction choice changes how much each trajectory contributes to the gradient, and thus substantially affects training dynamics and final results. The notation Seq-Tok-Tok denotes the reduction strategy for the policy / KL / NLL losses, respectively.
>
> | Reduction Strategy | Acc (%)  |   TPF    |    AUP    |
> | :----------------: | :------: | :------: | :-------: |
> |    Seq-Seq-Seq     |   87.5   |   5.42   |   467.1   |
> |    Seq-Tok-Seq     |   88.7   |   4.86   |   424.7   |
> |  **Seq-Tok-Tok**   | **90.3** | **5.58** | **492.4** |
> |    Tok-Tok-Tok     |   80.0   |   3.91   |   306.5   |
>
> ---
> **Cons2: Reward Formulation.**
>
> > The paper uses a joint reward over accuracy and TPF but has not provided the explicit reward formulation
>
> We thank the reviewer for pointing this out. We agree that the current draft does not state the reward components as explicitly as it should. In our implementation, each rollout $\tau^{(i,j)}$ is assigned two terminal signals: a correctness reward $r_{\text{acc}}^{(i,j)} = c^{(i,j)}$, where $c^{(i,j)}$ is the verifier-based correctness indicator, and a speed reward $r_{\text{tpf}}^{(i,j)} = \mathrm{TPF}(\tau^{(i,j)})$.
> LightningRL does not optimize a naive weighted sum of raw rewards. Instead, we first compute per-reward group-relative normalized advantages
>
> $$
> A_{\text{acc}}^{(i,j)} = \frac{r_{\text{acc}}^{(i,j)}-\mu_{\text{acc}}^{(i)}}{\sigma_{\text{acc}}^{(i)}+\epsilon}, \qquad
> A_{\text{tpf}}^{(i,j)} = \frac{r_{\text{tpf}}^{(i,j)}-\mu_{\text{tpf}}^{(i)}}{\sigma_{\text{tpf}}^{(i)}+\epsilon},
> $$
>
> aggregate them as
>
> $$
> A^{(i,j)} = A_{\text{acc}}^{(i,j)} + A_{\text{tpf}}^{(i,j)},
> $$
>
> and finally apply batch-level normalization, following Eq. 7 in the paper.
>
> We will revise Sec. 3.1 to make this formulation explicit.

---

> > ### Author Rebuttal · Reviewer_LDYx · 2026-04-03
> >
> > my questions have been adequately addressed. i remain my positive score.

---

> > > ### Author Response · Authors · 2026-04-05
> > >
> > > Thank you for your thoughtful feedback throughout the review process. We are glad that our clarifications have resolved your questions. Your initial insights were very helpful in refining our manuscript, and we truly appreciate your positive score and support.

---

### Official Review · Reviewer_93eY · 2026-03-18

**Soundness:** 3
**Presentation:** 2
**Significance:** 2
**Originality:** 2
**Overall Recommendation:** 3
**Confidence:** 4

**Summary:**

This paper focuses on improving the inference efficiency in diffusion models while maintaining accuracy. While existing dLLMs methods leverage parallel decoding, they trade off speed for accuracy, and either use specific decoding rules (e.g., uncertainty/entropy) or distill a faster decoding mechanism from a teacher. This paper proposes post-training diffusion LLMs with RL so that trajectories are both correct and efficiently parallelizable. Experiments on math and coding benchmarks show that accuracy, TPF, as well as a joint metric that captures the speed-accuracy trade-off, improve compared to baselines.

**Compliance With Llm Reviewing Policy:**

Affirmed.

**Key Questions For Authors:**

The RL stage has two rewards, one on accuracy and one on parallelism. Have you tried simulatenously optimize both of these rewards (i.e., Pareto improvement)?

**Limitations:**

Yes

**Strengths And Weaknesses:**

**Strengths:**
- The problem the paper studies is well-motivated and timely, given the recent interest in diffusion LLMs.
- The paper has extensive dataset, baseline, and model coverage in the Experiments section, showing consistent improvements over the baselines.

**Weaknesses:**
- The paper has limited novelty and is mainly a practical pipeline that introduces an extra parallel decoding reward into GRPO post-training of diffusion LLMs. The paper has a couple of simple training tricks, such as separately normalizing the correctness and parallelism reward components that help prevent collapse, and filtering based on whether sampled rollouts for a prompt show a difference in decoding speed (since GRPO learns by comparing rollouts within the same group). The token-level NLL anchor loss is widely used in LLMs and is not new.
- In principle, jointly optimizing for the difficult correctness reward and easier parallelism reward is reward hacking. One can also draw an analogy that adding a response length penalty constraint in RL post-training of the AR LLMs also tends not to work well.
- Writing can be improved, for example, some sentences in the abstract are hard to understand before reading the paper “improved training efficiency through dynamic sampling with TPF-aware filtering”.

---

> ### Author Rebuttal · Authors · 2026-03-31
>
> We sincerely thank the reviewer for the thoughtful comments and for acknowledging the clear motivation and extensive empirical evaluation of our work. We address your concerns below.
>
> ---
> **W1: Novelty.**
> >The reviewer questions the novelty of the paper, viewing LightningRL mainly as a practical GRPO-based post-training pipeline for dLLMs with an added parallel decoding reward. The reviewer also considers our key components to be simple or not fundamentally new.
>
> Thanks for the comments. While our detailed techniques draw inspiration from related work in the community, we clarify that the novelty of LightningRL lies in **formulating the breakdown of the accuracy–parallelism trade-off in dLLM as an RL problem**. Prior work typically aims to maximize parallelism across all possible decoding trajectories, whereas we argue that only the most promising candidates should be activated.
>
> To this end, we make substantial technical contributions beyond a straightforward application of existing RL methods to dLLMs: we identify new failure modes and design a principled multi-objective optimization framework. In contrast, naively joint optimization of correctness and parallelism tends to let the easier speed objective dominate, destabilizing training and degrading generation quality (e.g., Fig. 6a–6c).
>
> We will revise the paper to emphasize more clearly that our contribution does not reside in any isolated atomic component, but in the end-to-end, well-motivated optimization recipe. This is directly validated by our extensive ablation studies (e.g., Table 3 and Figs. 3–5), which show that removing key components leads to severe objective collapse.
>
> ---
> **W2: Reward Hacking.**
> >In principle, jointly optimizing for the difficult correctness reward and easier parallelism reward is reward hacking. One can also draw an analogy that adding a response length penalty constraint in RL post-training of the AR LLMs also tends not to work well.
>
> We agree that naively jointly optimizing a hard correctness reward and an easier parallelism reward can lead to fast-but-incorrect behavior. This is also the failure mode we observed when directly applying existing dLLM RL pipelines in this multi-objective setting: the parallelism objective can dominate the update, causing policy drift and degraded generation quality (e.g., Fig. 6a–6c).
>
> LightningRL addresses such a potential issue with per-reward decoupled normalization and token-level NLL anchoring on correct trajectories. As a result, instead of maintaining speed gains by sacrificing correctness, LightningRL improves the average AUP of SDAR-8B-b32 from **189.2 to 497.9** and the average TPF from **3.12 to 7.32** while keeping the average accuracy essentially unchanged (**71.0% vs. 71.1%**). Moreover, Fig. 6 shows that, unlike TraceRL under the same reward setting, LightningRL does not exhibit the same erosion of the accuracy signal during training.
>
> This is also consistent with the ablation results in Table 3: removing the proposed stabilizing components worsens both accuracy and TPF, rather than improving speed by sacrificing correctness.
>
> ---
> **W3: Writing.**
> >Writing can be improved, for example, some sentences in the abstract are hard to understand before reading the paper “improved training efficiency through dynamic sampling with TPF-aware filtering”.
>
> We thank the reviewer for pointing this out. In the revision, we will simplify this wording and state the intuition first. For example, instead of “improved training efficiency through dynamic sampling with TPF-aware filtering,” we will use phrasing such as “improves RL training efficiency by filtering out prompts with low parallelism contrast in sampling trajectories,” and introduce the technical term later in the main text. This should make the role of the mechanism clearer on first reading.
>
> ---
> **Q1: Pareto Improvement.**
> >The RL stage has two rewards, one on accuracy and one on parallelism. Have you tried simulatenously optimize both of these rewards (i.e., Pareto improvement)?
>
> Yes. LightningRL already jointly optimizes accuracy and parallelism rewards, with the two components normalized separately before combination. Empirically, we observe that during training, both accuracy and TPF improve together at first, showing clear Pareto improvement over the baseline. As training continues, the model continues to gain parallelism, while accuracy may fall back slightly from its peak. Even so, the overall Pareto frontier is still pushed outward. For example, on GSM8K, an intermediate checkpoint achieves **92.3%** accuracy and **4.25** TPF, compared with **88.9%** and **2.85** TPF for SDAR. We report the final checkpoint in the paper because it reflects a different operating point: higher parallelism with near-parity accuracy.

---

> > ### Author Rebuttal · Reviewer_93eY · 2026-04-02
> >
> > I thank the authors for their detailed response to my questions. However, my concerns regarding novelty and reward hacking (fundamental to the formulation and method) remain, and thus I maintain my score.

---

> > > ### Author Response · Authors · 2026-04-05
> > >
> > > Thank you for the follow-up. We appreciate the reviewer’s constructive feedback. We would like to make further clarifications on novelty and reward hacking.
> > >
> > > ---
> > > **Novelty**
> > >
> > > We emphasize that maximizing the inference speed of block-wise dLLMs without compromising output quality (i.e., breaking the accuracy-parallelism trade-off) is pretty vital in practice and has not been well accomplished by prior studies. LightningRL addresses this challenge from an RL post-training perspective, directly pushing forward the speed-quality Pareto frontier of existing dynamic large language models. This stands in clear contrast to recent approaches focused on dLLM distillation, and is empirically more effective.
> > >
> > > Besides, direct adaptations of prior RL methods (with the exact same accuracy and parallelism rewards) fail to match LightningRL's speed-quality frontier across benchmarks, as detailed below:
> > > |         Method         |  GSM8K (Acc/TPF/AUP)  | MATH500 (Acc/TPF/AUP) |   MBPP (Acc/TPF/AUP)    | HumanEval (Acc/TPF/AUP) |
> > > | :--------------------: | :-------------------: | :-------------------: | :---------------------: | :---------------------: |
> > > |      Couple-GRPO       |   75.3/ 4.22/ 254.7   |   59.1/ 4.93/ 283.1   |    54.0/ 2.61/ 139.9    |    68.2/ 2.42/ 164.6    |
> > > |        TraceRL         |   76.9/ 5.04/ 378.6   |   60.6/ 4.82/ 284.6   |    57.8/ 2.50/ 144.2    |    75.0/ 2.29/ 171.6    |
> > > |          DiRL          |   86.6/ 4.87/ 414.1   |   61.6/ 5.04/ 301.2   |    56.8/ 2.70/ 152.7    |    76.0/ 2.33/ 176.7    |
> > > | **LightningRL (Ours)** | **90.3/ 5.58/ 492.4** | **63.0/ 6.28/ 407.5** | **58.3/ 11.10 / 641.6** |  **72.6/ 6.30/ 450.1**  |
> > >
> > > It means that simply adding a parallelism reward to an existing dLLM RL pipeline is not sufficient to push the speed–quality frontier under this multi-objective setting.
> > >
> > > Moreover, we would like to mention the consensus among the other reviewers, who recognized our post-training RL formulation as a "natural and effective" (Reviewer b5jb) and a "new angle" (Reviewer LDYx) for tackling what is considered a "genuine bottleneck" in practical deployment. Our work has been recognized by Reviewer TXxQ for successfully addressing a critical gap in dLLM acceleration beyond existing training-free methods. We sincerely hope the reviewer will take these points into consideration and reevaluate the contributions of our work.
> > >
> > > ---
> > > **Reward Hacking**
> > >
> > > We respectfully clarify that reward hacking is primarily an empirical risk rather than something fundamentally unique to our formulation. In the broader LLM RL, whenever one objective is easier to optimize than another, there is always a risk that optimization may drift toward the easier signal.
> > >
> > > In fact, during our initial explorations, we empirically observed that naively combining correctness and parallelism rewards consistently led to optimization drift—the policy would exploit the easier parallelism signal at the expense of generation quality.
> > >
> > > Our ablation studies confirm that our proposed components are not cosmetic, but act as crucial regularizers against reward hacking.  For each variant, the results below are taken from the checkpoint that achieves the highest AUP within the first 20 training epochs on GSM8K train split:
> > > |            Case             | Acc (%) | TPF  |  AUP  |
> > > | :-------------------------: | :-----: | :--: | :---: |
> > > |        w/o NLL loss         |  80.7   | 5.03 | 385.7 |
> > > | w/o Decoupled normalization |  85.3   | 4.96 | 416.5 |
> > > |   w/o TPF-aware filtering   |  87.2   | 5.27 | 454.5 |
> > > |       Lightning(Full)       |  90.3   | 5.58 | 492.4 |
> > >
> > > As shown above, removing any of the three components consistently degrades the final operating point. This suggests that the gains do not come from simply pushing the easier speed reward; rather, per-reward decoupled normalization, token-level NLL anchoring, and TPF-aware filtering are all important for preventing drift and maintaining a better speed–quality frontier.
> > >
> > > Most importantly, the final evaluation does not show the pattern we would expect from reward hacking. Compared to SDAR-8B-b32, LightningRL increases the average TPF from 3.12 to 7.32 and the average AUP from 189.2 to 497.9 while keeping the average accuracy essentially unchanged (71.0% vs. 71.1%). On GSM8K, it even improves both accuracy and parallelism simultaneously, from 88.9 / 2.85 to 90.3 / 5.58 (Acc / TPF).
> > >
> > > ---
> > > We appreciate the reviewer’s time and effort in reconsidering these crucial aspects. To summarize, the core novelty of our work lies in both the formulation and the resolution of this bottleneck: identifying the accuracy-parallelism trade-off as an RL optimization challenge and solving it through a carefully adapted suite of well-motivated, well-studied techniques. Empirically, this principled design mitigates reward hacking and achieves a new SOTA speed-quality frontier among existing dLLMs. Thank you again for your valuable feedback, and we will continue improving the clarity and impact of our work.

---

### Decision · Program_Chairs · 2026-04-30

**Decision:**

Accept (regular)

**Comment:**

This paper addresses a critical bottleneck in diffusion-based Large Language Models: the trade-off between inference parallelism and task accuracy. The authors propose LightningRL, a post-training reinforcement learning framework designed to optimize the speed-quality Pareto frontier. The reviewers were initially divided, with two recommending acceptance and two recommending rejection. Reviewer LDYx and Reviewer b5jb were satisfied by the author's response, noting that the provided sensitivity analyses and wall-clock speed measurements successfully demonstrated the practical value and robustness of the method. Reviewer 93eY and Reviewer TXxQ maintained their negative recommendations, citing concerns over limited novelty, potential reward hacking, and incomplete ablation studies. After a careful reading of the reviews and the extensive rebuttal, I find the arguments for acceptance more persuasive. The authors addressed the "reward hacking" concern by providing empirical evidence that their specific components—such as decoupled normalization and NLL anchoring—effectively stabilize training and prevent accuracy degradation even as parallelism increases. Furthermore, the rebuttal significantly strengthened the paper by including comparisons against recent SOTA baselines like Couple-GRPO and DiRL, as well as providing concrete wall-clock latency improvements on H100 hardware. While Reviewer TXxQ remained concerned about experimental completeness, the authors' second-round clarifications on hyperparameters and advantage-signal stability provide a level of detail that meets the standards for a technically solid contribution. Given that the paper provides a natural and effective solution to a genuine deployment bottleneck and demonstrates consistent gains across multiple model scales, I recommend the paper for acceptance.